# AN EXAMINATION OF PREFERENCE-BASED REINFORCEMENT LEARNING FOR TREATMENT RECOMMENDATION

## ABSTRACT

Treatment recommendation is a complex multi-faceted problem with many conflicting objectives, e.g., optimizing the survival rate (or expected lifetime), mitigating negative impacts, reducing financial expenses and time costs, avoiding over-treatment, etc. While this complicates the hand-engineering of a reward function for learning treatment policies, fortunately, qualitative feedback from human experts is readily available and can be easily exploited. Since direct estimation of rewards via inverse reinforcement learning is a challenging task and requires the existence of an optimal human policy, the field of treatment recommendation has recently witnessed the development of the preference-based Reinforcement Learning (PRL) framework, which infers a reward function from only qualitative and imperfect human feedback to ensure that a human expert's preferred policy has a higher expected return over a less preferred policy. In this paper, we first present an open simulation platform to model the progression of two diseases, namely Cancer and Sepsis, and the reactions of the affected individuals to the received treatment. Secondly, we investigate important problems in adopting preference-based RL approaches for treatment recommendation, such as advantages of learning from preference over hand-engineered reward, addressing incomparable policies, reward interpretability, and agent design via simulated experiments. The designed simulation platform and insights obtained for preference-based RL approaches are beneficial for achieving the right trade-off between various human objectives during treatment recommendation.

## 1    INTRODUCTION

With recent advances in deep learning and open access to large-scale Electronic Health Records (EHRs), Deep Reinforcement Learning (RL) approaches have gained popularity for treatment recommendation (Raghu et al., 2017; Lopez-Martinez et al., 2019). But the success of RL applications often crucially depends on the prior knowledge that goes into the definition of the reward function (Wirth et al., 2017). However, treatment recommendation is a multi-faceted problem where the reward function is hard to engineer and requires quantifying the trade-off between diverse types of realistic objectives. For instance, clinicians often aim to optimize the survival rate (or expected lifetime) while mitigating negative impacts of the treatment (Raghu et al., 2017; Lopez-Martinez et al., 2019; Wang et al., 2018). However, they also keep in mind, the patient's considerations of financial expenses and time costs in accepting treatment strategies (Faissol et al., 2007; Denton et al., 2009). Moreover, unnecessary or over-treatment needs to be avoided and certain agreements based on the patient's medical insurance plan also need to be followed for an affordable treatment (Nemati et al., 2016).

To explicitly reflect human's objectives in the reward function, prior work jointly considers multiple objectives weighted linearly to reduce the problem to a single-objective MDP (Faissol et al., 2007; Denton et al., 2009). However, the linearly weighted reward function induces negative interference between objectives, especially when representations are learned using neural networks and shared among different objectives, which goes against human's actual intentions (Pham et al., 2018; Schaul et al., 2019). Further, given clinicians' treatment strategies, the intrinsic reward function cannot be inferred accurately with existing inverse reinforcement learning (IRL) methods (Abbeel & Ng,

2004; Ho & Ermon, 2016), since they require access to samples from an optimal policy, which is not guaranteed in reality (Komorowski et al., 2018; Saria, 2018).

Fortunately, qualitative feedback according to humans' preferences can be easily obtained and efficiently leveraged to infer reward functions. In this paper, we investigate preference-based Reinforcement Learning approaches (Fürnkranz et al., 2012; Cheng et al., 2011; Akrour et al., 2012; Schäfer & Hüllermeier, 2018; Christiano et al., 2017) for treatment recommendation, where the reward is estimated based on preferences over a pair of treatment strategies. Specifically, the reward estimator ensures that in a policy pair, the policy preferred according to a human's objectives has a higher expected return. However, acceptance of PRL approaches for treatment recommendation requires significant exploration of their practical utility, reliability and interpretability.

**Contributions**: In this work, we first present an open simulation platform to investigate the preference-based Reinforcement Learning approaches from the above aspects. The constructed simulator models the dynamic state transitions of different individuals with Cancer or Sepsis and their reactions to the received medication or operation treatment, which enables efficient model training and reliable performance evaluation. Next, we conduct comprehensive simulated experiments to address the following questions: 1) Does the preference-based qualitative feedback really benefit the policy learning compared to handcrafted rewards and other existing treatment recommendation approaches? 2) How to better optimize human's objectives by learning a reward representation which can deal with policies which are incomparable? 3) Is the reward function inferred by PRL interpretable and does it faithfully follow human intentions? and 4) How to design agent types so that resulting policies together with the preference feedback lead to more accurate reward estimation and better treatment outcomes? Our experiments provide useful insights and guidance in developing preference-based RL approaches to realize the right trade-off between human objectives during treatment recommendation.

## 2 PROBLEM DEFINITION

We cast the treatment policy learning as a Markov Decision Process (MDP). At time-step $t$, $s_t$ is a vector composed of multiple health-related features, $a_t$ is either a scalar value representing dosage amount or a boolean value denoting whether to perform an operation. Besides effects from the conducted actions, features in the state influence each other's progression, which is simulated by the state transition probability function $\mathcal{P}(s_{t+1}|s_t, a_t)$. The agent is targeted at learning the optimal policy $\pi^*$ that maximizes the expected return $\mathcal{V}_{\pi^*}(s_0) = \max_\pi \mathbb{E}[\sum_{t=0}^\infty \gamma^t r_t]$, where $\gamma \in [0, 1]$ is the discount factor and $r_t$ is the estimated reward based on preference feedback. Given two policies $\pi_m$ and $\pi_n$ starting with the same initial state $s_i$, $\pi_m(s_i) \succ \pi_n(s_i)$ represents the preference of policy $\pi_m$ to $\pi_n$ based on human's objectives. Rather than using hand-crafted reward functions of the MDP, we are aimed at finding a parameterized reward function $r_{\theta_P}$ that approximates the true reward function $r$ underlying human's preference.

## 3 SIMULATION PLATFORM DESIGN

### 3.1 GENERAL CANCER AND DRUG TREATMENT SIMULATION

Following prior work (Fürnkranz et al., 2012), we use the mathematical model proposed by Zhao et al. (2009) to simulate the general cancer evolution and drug treatment effects.

**State Transformation:** The values of the next tumor size $y_{t+1}$ and the toxicity level $x_{t+1}$ are determined by the current drug amount $d_t$, their current values $y_t, x_t$ and initial values $y_0, x_0$:

$$y_{t+1} = \text{ReLU}\big(y_t + [a_1 \cdot \max(x_t, x_0) - b_1 \cdot (d_t - m_1)] \times I(y_t > 0)\big)$$
$$x_{t+1} = \text{ReLU}\big(x_t + a_2 \cdot \max(y_t, y_0) + b_2 \cdot (d_t - m_2)\big),$$

where $I$ is the indicator function which outputs 1 if the current tumor size $y_t > 0$ and 0 otherwise.

### 3.2 SEPSIS INFECTION AND BLOOD PURIFICATION SIMULATION

We employ the mathematical model derived by Song et al. (2012) to simulate the acute inflammation process in response to an infection. There are 19 physiological features that govern sepsis dynamics,

8 of which are observable while the remaining 11 are unmeasurable conceptual variables. Whenever a blood purification operation is made, three components in the circulation are eliminated, i.e., activated neutrophils $N_a$ and the pro- and anti-inflammatory mediators *PI* and *AI*. Besides effects from the blood purification operation, the variables influence each others' progression through Ordinary differential equations (ODEs).

**State Transformation:** There are 18 ODEs to describe feature interactions and 3 ODEs for hypothetic mechanism of blood purification. For a simple demonstration, we only list the equations of activated neutrophils ($N_a$) with(out) blood purification operation here.

$$\text{No operation: } \frac{dN_a}{dt} = \frac{N_r PI^n}{h^n_{N_r\text{-}N_a} + PI^n} \frac{1}{\tau_{N_r\text{-}N_a}} + \frac{N_p PI^n}{h^n_{N_p\text{-}N_a} + PI^n} \frac{1}{\tau_{N_p\text{-}N_a}} - \frac{N_a}{\tau_{N_a}}$$
$$- \frac{N_a PI^n}{h^n_{N_a\text{-}N_s} + PI^n} \frac{1}{\tau_{N_a\text{-}N_s}},$$
$$\text{With operation: } \frac{dN_{aHA}}{dt} = \frac{dN_a}{dt} - \frac{N_a/N_\infty}{h_{AIHA} + (N_a/N_\infty)},$$

where $N_r, N_p, N_a$ are resting, primed and activated blood neutrophils respectively, *PI* is the systemic pro-inflammatory response, $\tau_{N_r\text{-}N_a}, \tau_{N_p\text{-}N_a}, \tau_{N_a\text{-}N_s}$ are constant parameters and $h^n_{N_r\text{-}N_a}, h^n_{N_p\text{-}N_a}, h^n_{N_a\text{-}N_s}, h_{AIHA}$ are hill equations.

# 4 METHOD

## 4.1 EXISTING TREATMENT RECOMMENDATION APPROACHES

**Learning from Hand-crafted Reward:** When the optimization objective is to maximize the clinical efficacy alone, the reward function for intermediate timesteps is either 0 or hand-crafted based on indicators of patient health, while the rewards for positive and negative outcomes at terminal timesteps are normally of the same scale, but of opposite directions (Raghu et al., 2017; Wang et al., 2018; Nemati et al., 2016). Besides obtaining the optimal clinical efficacy, another line of work has also included auxiliary objectives like mitigating negative impacts or improving health conditions. Most of existing work specified linear scalarization functions based on domain knowledge to project the multi-objective MDP to a single-objective MDP (Denton et al., 2009; Lopez-Martinez et al., 2019; Zhao et al., 2009). Due to the sensitivity of the learned policy and resulted performance to relative values of the manually specified rewards, the employed reward functions in these approaches are difficult to be quantified by experts to achieved distinct goals in treatment recommendation.

**Learning from Human Feedback:** Given demonstrations from domain experts, inverse Reinforcement Learning (IRL) methods have been proposed to seek the reward function that models the intention of the demonstrator first and then train RL agents to match the demonstrations (Abbeel & Ng, 2004; Ho & Ermon, 2016). Though explicit quantitative reward signals are no longer needed in IRL settings, learning treatment policies from clinicians' demonstrations is challenging, since optimal demonstrations are difficult to provide by clinicians while the general treatment regimens in demonstrations can hardly reflect the actual intentions (Gao et al., 2018; Brown et al., 2019). Fortunately, even non-experts can provide feedback in the form of preference, which has been utilized to replace conventional numerical reward signals with relative utility values (Fürnkranz et al., 2012; Cheng et al., 2011; Akrour et al., 2012; Schäfer & Hüllermeier, 2018; Christiano et al., 2017).

## 4.2 PREFENCE-BASED REINFORCEMENT LEARNING FRAMEWORK

We display the framework in Algorithm 1 to show the reward and policy learning procedure given preference feedback. Firstly, we are aimed at learning a reward function, based on which the preference between two policies could be approximated.

**Learning from Qualitative Feedbacks:** We denote by $\pi_m(s_i) \succ \pi_n(s_i)$ the case that given $s_i$, $\pi_m$ is preferred to $\pi_n$. We here treat the qualitative feedback learning problem as a classic binary classification task, where two policies are given and a model learns to approximate the preference between the two. We assume the probability that one policy is preferred to the other is a function of their received reward estimations (explained later in *Preference Probability Representation*), then the

---

**Algorithm 1** PREFERENCE-BASED RL FOR TREATMENT RECOMMENDATION

---

**Require:**
    $S'$: initial states of sampled subjects
    $N$: number of training iterations
    $T$: the maximum simulation time to treat each subject
1: Randomly initialize $\theta_P, \theta_A^1, \theta_A^2$
2: $\mathcal{D} = \emptyset, \Gamma^1 = \emptyset, \Gamma^2 = \emptyset$ // Initialize empty lists to store samples for reward and agent learning
3: **for** $n = 0$ **to** $N - 1$ **do**
4:     **for all** $s \in S'$ **do**
5:         $s_0^1 \leftarrow s, s_0^2 \leftarrow s, \tau^1 \leftarrow \emptyset, \tau^2 \leftarrow \emptyset$ // Initialize state vector and trajectory list
6:         **for** $t = 0$ **to** $T - 1$ **do**
7:             $a_t^1 \leftarrow \pi(s_t^1; \theta_A^1), s_{t+1}^1 \leftarrow \text{SIMULATE}(s_t^1, a_t^1), r_{\theta_P, t}^1 \leftarrow \text{REWARD}(s_t^1, a_t^1; \theta_P)$
8:             $a_t^2 \leftarrow \pi(s_t^2; \theta_A^2), s_{t+1}^2 \leftarrow \text{SIMULATE}(s_t^2, a_t^2), r_{\theta_P, t}^2 \leftarrow \text{REWARD}(s_t^2, a_t^2; \theta_P)$
9:             $\tau^1 \leftarrow \tau^1 \cup \{(s_t^1, a_t^1, r_{\theta_P, t}^1), s_{t+1}^1\}, \tau^2 \leftarrow \tau^2 \cup \{(s_t^2, a_t^2, r_{\theta_P, t}^2, s_{t+1}^2)\}$
10:         **end for**
11:         $\Gamma^1 \leftarrow \Gamma^1 \cup \{\tau^1\}, \Gamma^2 \leftarrow \Gamma^2 \cup \{\tau^2\}$
12:         $pre(\tau^1, \tau^2) \leftarrow \text{EVALUATEPREFERENCE}(\tau^1, \tau^2)$ // Preference feedback from humans
13:         $\mathcal{D} \leftarrow \mathcal{D} \cup (\tau^1, \tau^2, pre(\tau^1, \tau^2))$
14:     **end for**
15:     Drawing minibatches $\Gamma_n^1 \sim \Gamma^1, \Gamma_n^2 \sim \Gamma^2, \mathcal{D}_n \sim \mathcal{D}$
16:     Update reward network $\theta_P$ with $\mathcal{D}_n$ by PREFERENCE-BASED REWARD LEARNING// See Algorithm 2 (**Appendix**)
17:     Update agent network $\theta_A^1$ with $\Gamma_n^1$, $\theta_A^2$ with $\Gamma_n^2$ by PREFERENCE-GUIDED AGENT LEARNING// See Algorithm 3 (**Appendix**)
18: **end for**

---

cross-entropy loss function is:

$$L(\theta_P) = -\mathbb{E}_{s_i \sim S}\big[I\big(\pi_m(s_i) \succ \pi_n(s_i)\big) \log p\big(\pi_m(s_i) \succ \pi_n(s_i); \theta_P\big) \tag{1}$$
$$+ I\big(\pi_n(s_i) \succ \pi_m(s_i)\big) \log p\big(\pi_n(s_i) \succ \pi_m(s_i); \theta_P\big)\big],$$

where $I(\cdot \succ \cdot)$ is an indicator function equal to 1 if the first policy is preferred to the second, 0 otherwise.

**Preference Probability Representation:** Bradley–Terry model (Bradley & Terry, 1952) is a widely used probability model to predict the preference of a paired comparison: $p(i \succ j) = \frac{p_i}{p_i + p_j}$, where $p_i$ is a positive real-valued score assigned to individual $i$. In order to compute the probability that $\pi_m$ is preferred to $\pi_n$ given state $s_i$, we employ its implementation introduced in (Agresti & Kateri, 2011) :

$$p\big(\pi_m(s_i) \succ \pi_n(s_i)\big) = \frac{\exp R(\pi_m, s_i; \theta_P)}{\exp R(\pi_m, s_i; \theta_P) + \exp R(\pi_n, s_i; \theta_P)}, \tag{2}$$

where capital $R$ denotes the expected return of conducting a policy given one specific initial state.

Given the learned reward, parameters of the RL agent are updated with either of the following two methods.

**Action-based Reward Modification (AbRM):** Hand-crafted rewards assigned to different outcomes influence the agent performance a lot, even if we keep the ratio but change the magnitude only. Instead of designing the scalar rewards for different outcomes manually, we send the preference-based reward $r_{\theta_P}(s_t, a_t)$ to the agent at each time-step.

**State-based Reward Modification (SbRM):** We derive a new state value $h_{\theta_P}$ from $r_{\theta_P}$ to represent how good the state is: $h_{\theta_P}(s_t) = \max_a r_{\theta_P}(s_t, a)$. We further compute the advantage value of the current state over the previous one, $h_{\theta_P}(s_t) - h_{\theta_P}(s_{t-1})$, as the instant reward to encourage appropriate behaviors in accordance with preference.

**Problems to be Resolved:** Before applying preference-based RL approaches to treatment recommendation, we need to address the following problems to ensure that the preference-based reward estimation is consistent with human's objectives and the well-trained agent provides reliable and interpretable policies to clinicians:

Table 1: Performance for Cancer medication recommendations. The best result per metric is marked in boldface. We present $avg \pm stdev$ values for all experiments averaged over 10 independent runs.

| Method Type | Method Name | Clinical Efficacy | Other Factors |
|---|---|---|---|
| | | Survival Rate | Tumor + Toxicity |
| Non-learning | Constant Best (0.4) | 19.91%±0.58% | 2.22±0.04 |
| | Constant Worst (0.1) | 4.89%±0.68% | 3.72±0.03 |
| | Random | 17.81%±0.91% | 2.23±0.04 |
| Preference Learning | PBPI | 21.79%±0.64% | 2.21±0.07 |
| Reinforcement Learning (handcrafted reward) | Single-objective RL | 26.96%±3.02% | 1.16±0.48 |
| | Single-objective RL (*Ensemble*) | 27.38%±3.32% | 1.14±0.49 |
| | Existing Multi-objective RL | 18.84%±5.77% | 2.28±0.66 |
| | Grid-search Multi-objective RL | 28.98%±3.42% | 0.66±0.45 |
| Reinforcement Learning (Preference-based reward) | AbRM (*CE*) | 31.52%±1.38% | 0.46±0.06 |
| | AbRM (*CE & OF-I*) | 31.33%±1.18% | **0.39±0.02** |
| | SbRM (*CE*) | 30.54%±3.46% | 0.68±0.45 |
| | SbRM (*CE&OF-I*) | **31.72%±1.08%** | 0.43±0.06 |

- Does the preference-based qualitative feedback really benefit the policy learning compared to handcrafted rewards and other existing treatment recommendation approaches?
- How to better optimize human's objectives by learning a reward representation which can deal with policies which are incomparable?
- Is the reward function inferred by PRL interpretable and does it faithfully follow human intentions?
- How to design agent types so that resulting policies together with the preference feedback lead to more accurate reward estimation and better treatment outcomes?

## 5 EXPERIMENTS

### 5.1 SETTINGS

**Medication Recommendation for General Cancer:** For 6-month simulation, the agent makes dosage amount decisions in each month. $10,000$ subjects are randomly sampled for training, $2,000$ for validation and $2,000$ for testing. We are aimed at learning optimal policies with three kinds of intentions: 1) maximizing survival rate to obtain optimal clinical efficacy (*CE*); 2) and mitigating negative effects represented by the sum of the tumor size and the toxicity level after treatment (*CE&OF-I*); 3) and mitigating negative effects represented by two separate health signs, the highest toxicity level during the treatment and the final tumor size (*CE&OF-II*).

**Blood Purification Recommendation for Sepsis:** During the 100-hour simulation, the agent is asked whether to perform a 2-hour operation in every 2 hours. We randomly sample $3,000$ subjects for training, $1,000$ for validation and testing. This is a partially observable MDP and LSTMs are utilized for agent modeling. The agent learns policies to fulfill two intentions: 1) maximizing survival rate to obtain optimal clinical efficacy (*CE*); 2) and avoiding too frequent operations (*CE&OF*).

### 5.2 COMPARED APPROACHES

We benchmark results from the following existing approaches from treatment recommendation literature:
- Non-learning (Zhao et al., 2009; Fürnkranz et al., 2012): 1) *Constant*: A static dosage amount is given to all the subjects throughout the six months; 2) *Random*: One of the four dosage options is randomly selected at each time-step; 3) *Upper Bound*: The subjects with Sepsis receive operations all the time throughout the simulation period.
- Preference Learning (Fürnkranz et al., 2012): in Preference-Based Policy Iteration (*PBPI*), one action is preferred to the other based on their outcomes after certain times of simulations. Every time the dosage with the highest preference is selected.
- Reinforcement Learning with handcrafted Reward: 1) *Single-objective RL* (Schulman et al., 2015): the conventional policy gradient approach; it receives $+1$ for survival outcome, $-1$ for death, and $0$ for all intermediate steps. 2) *Single-objective RL (Ensemble)*: among two agents, the one with better performance on the validation set is evaluated on the testing set. It is developed for fair comparison

Figure 1: Performance for Sepsis blood purification recommendation.

(a) Optimize clinical efficacy only

(b) Optimize clinical efficacy & mitigate negative impacts

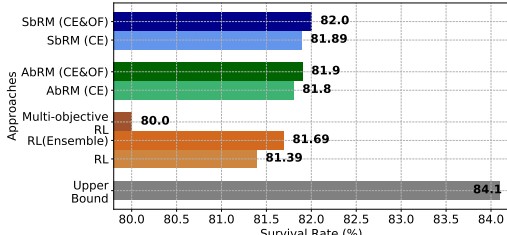
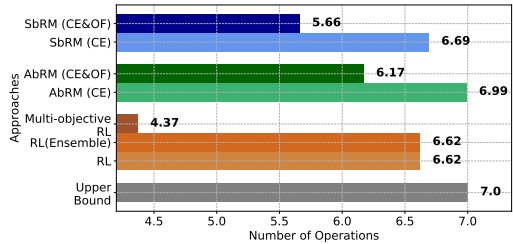

as two agents with different parameter initializations are used in the PRL framework. 3) *Existing Multi-objective RL* (Zhao et al., 2009): manually defined rewards based on key factors are assigned to each time-step. 4) *Grid-search Multi-objective RL* (Kim & De Weck, 2006): both the clinical efficacy and the negative impacts from other factors are treated as objectives and the linear scalarization with the best weight retrieved from grid search is employed.

• Reinforcement Learning with Preference-based Reward: to guide the RL agent learning, both *AbRM* and *SbRM* are trained based on preference determined by human's intentions. Since the preference-based reward is a non-stationary value approximated by a neural network, we implement agents with the policy gradient, which is robust to changes in the reward function (Ho & Ermon, 2016; Christiano et al., 2017).

## 5.3 BENCHMARK RESULTS

**Medication Recommendation for General Cancer:** We evaluate different approaches to pursue treatment goals in terms of maximizing survival rate *CE*, mitigating negative impact *CE&OF-I* (sum of tumor size and toxicity level) in Table 1 and *CE&OF-II* (highest toxicity level and final tumor size) in Table 2 (**Appendix**). Considering the *Survival Rate* as the only metric to derive preference on two policies, agents learning from either action-based ($31.52\%$) or state-based ($30.54\%$) preference reward have much better performance in saving lives than *Single-objective RL* ($26.96\%$), where the handcrafted reward is used to penalize policies with death outcomes. When negative impacts are expected to be mitigated besides saving lives, agents receiving rewards from preference are capable to maintain the performance on the clinical efficacy with much fewer negative impacts at the same time. When human's preference is defined as *CE&OF-II*, the *highest toxicity level* during the treatment and the *final tumor size* are two contradictory objectives to minimize. In Table 2, we observe that preference-based reward guides the agent to policies with the highest survival rates, while one negative impact gets reduced but the other increases compared with other approaches.

**Blood Purification Recommendation for Sepsis:** Performance bar charts of different approaches evaluated by *Survival Rate* and *Number of Operations* are illustrated in Fig. 1. When guided by preference-based reward rather than manually crafted reward, a slightly higher *Survival Rate* is achieved by both *AbRM* and *SbRM*, while the average number of operations has fallen considerably, by $6.79\%$ with *AbRM* and $14.50\%$ with *SbRM*. Note that although the approach *Multi-objective RL* leads to the fewest number of operations, the performance in *Survival Rate* has dropped to make undesired trade-offs between clinical efficacy optimization and negative impacts mitigation.

## 5.4 EXAMINATION OF PREFERENCE-BASED RL APPROACHES

**Advantages of Preference-based Reward over handcrafted Reward:** In Fig. 5 (**Appendix**), we first show the sensitivity of policy behaviors to small changes in handcrafted rewards leads to unstable clinical efficacy even when the relative importance of obtaining positive outcomes against negative ones keeps unchanged. The three heatmaps of the agent's performance in response to different reward scalars reflect the difficulty in specifying an appropriate reward function to enable policy learning with the optimal clinical efficacy during treatment recommendation. In Fig. 6 (**Appendix**), we further show the difficulty of selecting reward scalars for three factors – survival rate, last tumor size and maximum toxicity level – in the grid-search Multi-objective RL approach to appropriately prioritize the clinical efficacy over negative impacts. To study whether the preference-based reward estimation can fully capture the human's intentions, we provide the RL agents with linear combinations of

Figure 2: Sepsis treatment strategies recommended by the SbRM agent: (a) clinical efficacy and expected return during training, (b) reward transferability among different configurations.

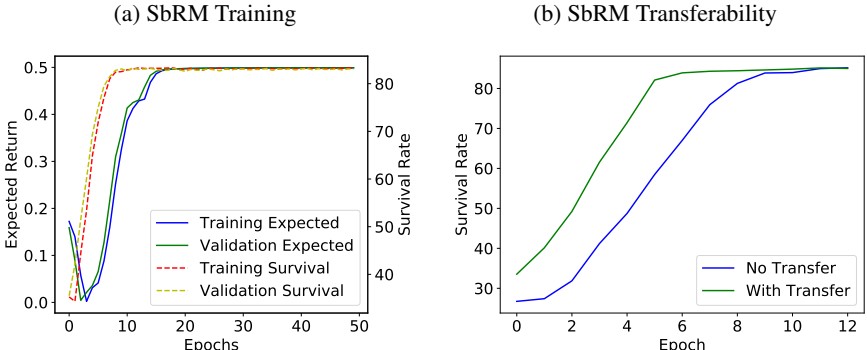

preference-based reward and handcrafted reward. Results in Table 3 (**Appendix**) show that learning from preference-based reward alone is adequate to achieve high clinical efficacy, while its combination with handcrafted rewards distracts the RL agent from optimizing the human's actual intentions and finally leads to inferior performance. To study the generalizability of the reward estimator, we extract the well-trained reward model in 2-hour operation configuration for Sepsis treatment and use it as the pre-trained model for 4-hour operation experiments. In Fig. 2b and Fig. 9b (**Appendix**), the reward estimator with knowledge transfer helps the agent speed up learning: compared with learning from scratch, the reward estimator with good initialization from a different configuration can provide better guidance to the agent.

Figure 3: Cancer treatment recommendation for 10,000 training subjects with the PRL framework. Curves in green describe scenarios when incomparable policies are discarded while curves in blue show cases when comparable policies are preserved and efficiently utilized.

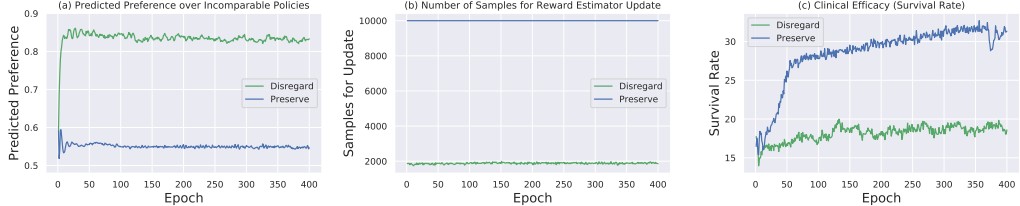

**Addressing Incomparable policies:** Given two policies for one sampled subject, if they have identical performance according to human's objectives, then the two policies are deemed to be incomparable. Since no clear preference conclusion can be drawn between the two incomparable policies, the majority of existing work in preference learning disregarded them directly (Fürnkranz et al., 2012; Cheng et al., 2011; Akrour et al., 2012; Schäfer & Hüllermeier, 2018; Christiano et al., 2017). Only comparable pairs, either $\pi_m$ preferred to $\pi_n$ $(I(\pi_m(s_i) \succ \pi_n(s_i)) = 1)$ or $\pi_n$ preferred to $\pi_m$ $(I(\pi_n(s_i) \succ \pi_m(s_i)) = 1)$, are included in the training set to optimize preference approximation. However, preference learning based on comparable policies alone achieves quite unsatisfactory clinical efficacy in our treatment recommendation tasks. As shown in Fig. 3c for Cancer treatment recommendation, the survival rate (green curve) progresses with little improvement but great fluctuation during 400 epochs of training. Two reasons are likely to contribute to the failure: 1) polarized preference (one preferred with probability 0.85, and the other 0.15 in Fig. 3a) is inferred between two incomparable policies although the preference label is never provided in the training set; 2) only around one-fifth of the policy pairs (2,000 comparable from 10,000 sampled subjects) are leveraged in each epoch for preference model update (green line in Fig. 3b). After the above performance analysis, we find that excluding incomparable pairs from the training set leaves the parameterized model exploring the preference space arbitrarily and inferring random preference over two policies although they are incomparable. To avoid arbitrary exploration in the preference space, we handle the incomparable pairs with a simple approach: treating both policies from the incomparable pair equally, i.e., $I(\pi_m(s_i) \succ \pi_n(s_i)) = I(\pi_n(s_i) \succ \pi_m(s_i)) = 0.5$. With the small but important augmentation to the preference indicator function, incomparable policies are efficiently

Figure 4: Cancer treatment recommendation: (a) clinical efficacy and expected return during training, and (b, c) expected return of policies ending with different negative impacts during testing.

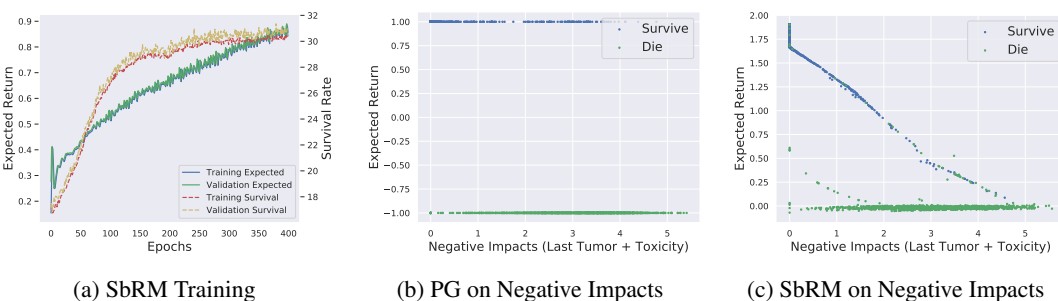

(a) SbRM Training  (b) PG on Negative Impacts  (c) SbRM on Negative Impacts

utilized for better preference space exploration (preference approaching 0.5 as expected in Fig. 3a), more samples for preference model update (all the 10,000 samples from the training set participate in the loss function minimization in Fig. 3b), and much higher clinical efficacy (more than 30% survival rate achieved after the model converges in Fig. 3c).

**Interpretability in Inferred Rewards:** To demonstrate whether the preference-based reward match human's actual intentions, we visualize the *SbRM* and *AbRM* agent's expected return for Cancer treatment during training and its relationship with the resulted negative impacts during testing (*OF-I*) in Fig. 4 and Fig. 7 (**Appendix**), respectively. From Fig. 4a, we can observe that the rising trend of the expected return matches the improving *Survival Rate* quite well, although the parameters of the reward estimator are updated at the same time. The estimated reward offers reasonable explanations for the policy performance: the higher the expected return, the better the policy. After the model converges, we analyze the distribution of expected returns for policies with different negative impacts. Since penalties or rewards are assigned to policies based on their outcomes only, the conventional Policy Gradient approach treats policies ending with survivals but different negative impacts equally (horizontal blue dots in Fig. 4b). After adopting preference-based reward, policies resulting in survival outcomes can distinguish from each other: policies with smaller negative impacts have much higher expected return. In Fig. 4c, policies leading to deaths have extremely low expected return (approaching zero), while the expected return for policies with survival outcomes is negatively proportional to the amount of negative impacts. As shown in Fig. 2a and Fig. 9a (**Appendix**), the expected return received by the agent for Sepsis treatment also shares the common trend with the *Survival Rate*: if more lives have been saved by the agent, then higher expected return is achieved.

**Influence of Agent Types in Treatment Outcomes:** As depicted in Algorithm 1, the studied preference-based RL framework adopts two RL agents controlled by different parameters to infer the reward and learn the policy that optimizes human's intentions. We here study the influence of different agent designs on reward approximation and resulted performance. Specifically, the reward function is estimated to approximate the preference over policies, among which the first policy is performed by one RL agent while the second policy can be executed by different agent types. The clinical efficacy curves shown in Fig. 8 (**Appendix**) empirically prove the effectiveness of the current design of two different preference-based RL agents.

## 6 CONCLUSIONS AND FUTURE DIRECTIONS

To obtain optimal treatment policies based on human's diverse objectives, we investigate performance of the preference-based Reinforcement Learning approaches, where higher rewards are automatically estimated and assigned to actions following human's actual intentions underlying the provided preference feedback. During interacting with the developed simulation platform, we resolve critical implementation problems and gain a deeper understanding in designing preference-based RL approaches, in order to better aid clinicians in treatment decision making. In future work, we will consider tackling some more practical aspects about human's preferences in adopting treatment strategies: 1) how to efficiently leverage preference in reward learning if human's feedback is limited, 2) how to fully reflect human's actual intentions in reward learning if both preference feedback and clinicians' demonstrations are provided.

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

# A  APPENDIX

## A.1  FIGURES AND TABLES

Figure 5: Performance of RL in different hand-crafted reward designs to optimize clinical efficacy.

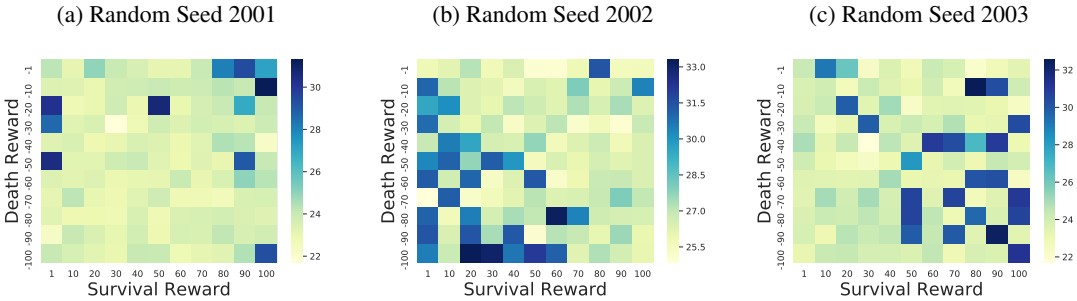

(a) Random Seed 2001  (b) Random Seed 2002  (c) Random Seed 2003

Figure 6: Performance of Multi-objective RL with linear scalarization on three reward factors: survival rate, last tumor size, and maximum toxicity levels. The linear weight assigned to each factor is one of four values: $\{1, 2, 4, 8\}$.

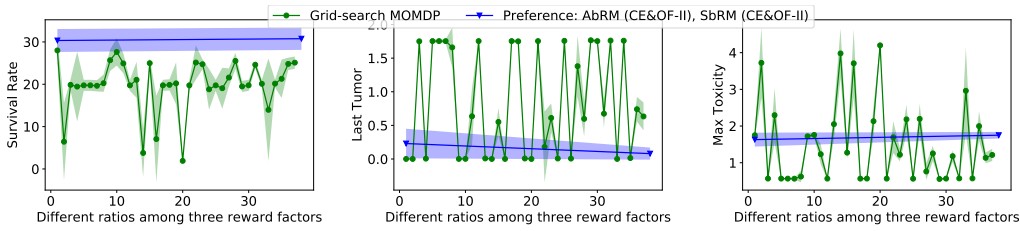

Figure 7: Cancer treatment strategies recommended by agent AbRM: (a) clinical efficacy and expected return during training, and (b) expected return of policies ending with different negative impacts during testing.

(a) AbRM Training  (b) AbRM on Negative Impacts

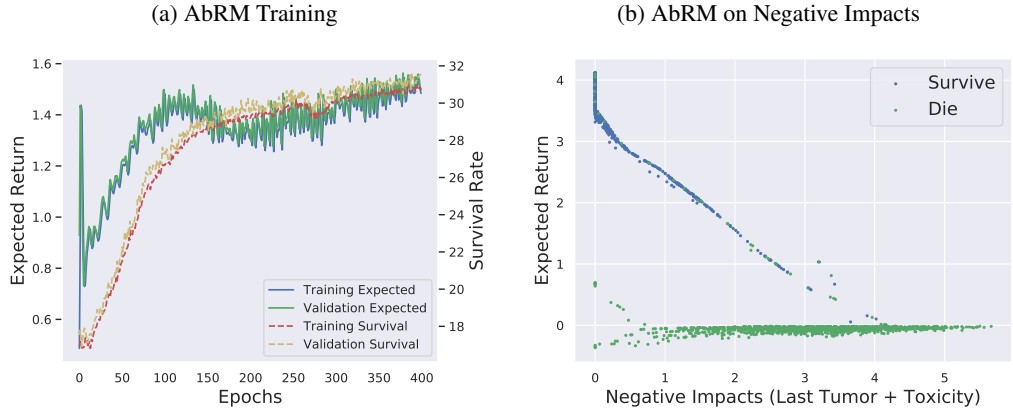

Figure 8: Effects of different agent designs on performance.

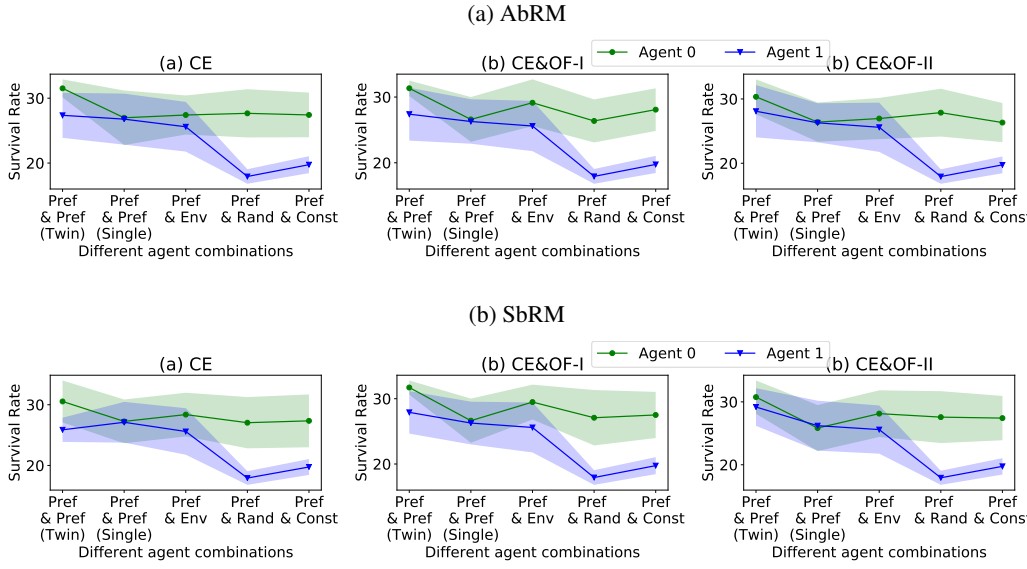

Figure 9: Sepsis treatment strategies recommended by the *AbRM* agent: (a) clinical efficacy and expected return during training, (b) reward transferability among different configurations.

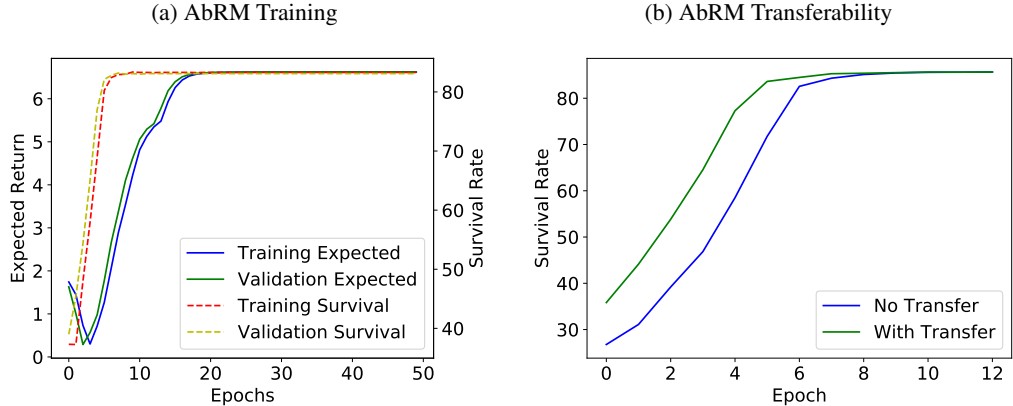

Figure 10: For Cancer experiments, true expected return of DQN learning from behavioral policies of Policy Gradient and its estimations from different off-policy evaluation methods.

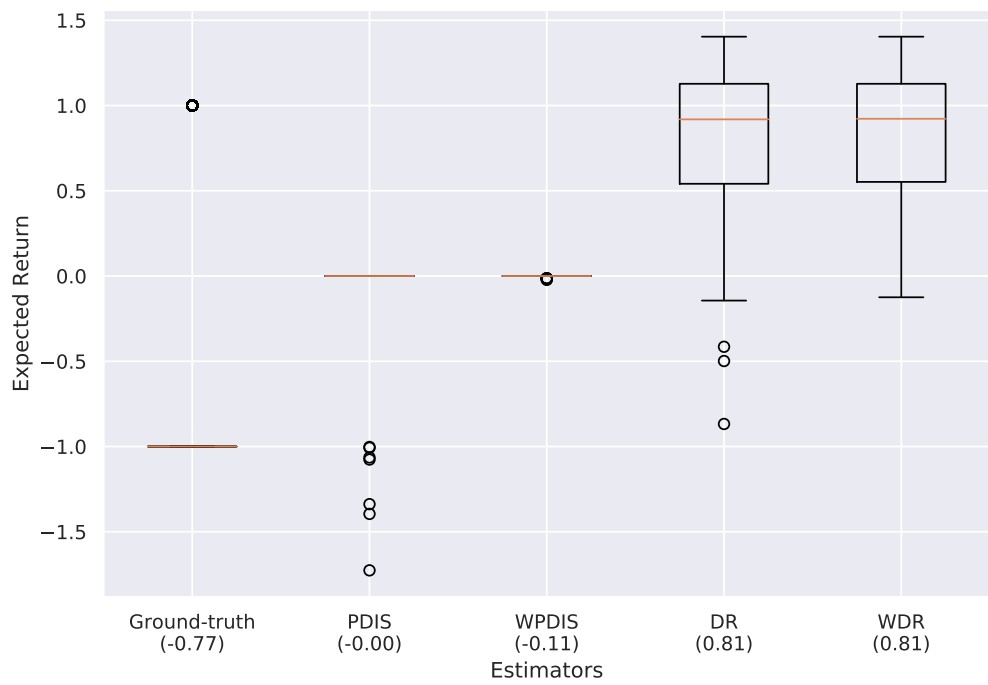

Table 2: Performance for Cancer medication recommendation considering negative impacts from two factors: the tumor size in the end and the ever experienced maximum toxicity.

| Method Type | Method Name | Clinical Efficacy | Other Factors | |
| --- | --- | --- | --- | --- |
| | | Survival Rate | Last Tumor | Max Toxicity |
| Non-learning | Constant Best (0.4) | 19.91%±0.58% | 1.76±0.02 | 0.57±0.01 |
| | Constant Worst (0.1) | 4.89%±0.68% | 3.72±0.03 | **0.48±0.04** |
| | Random | 17.81%±0.91% | 0.82±0.02 | 1.64±0.04 |
| Preference Learning | PBPI | 20.80%±0.56% | 1.38±0.12 | 1.01±0.12 |
| Reinforcement Learning (Hand-crafted reward) | Single-objective RL | 26.96%±3.02% | 0.48±0.28 | 1.37±0.28 |
| | Single-objective RL (*Ensemble*) | 27.38%±3.32% | 0.47±0.29 | 1.38±0.29 |
| | Existing Multi-objective RL | 18.84%±5.77% | 0.25±0.10 | 2.16±0.60 |
| | Grid-search Multi-objective RL | 25.10%±1.44% | 0.63±0.20 | 1.21±0.16 |
| Reinforcement Learning (Preference-based reward) | AbRM (*CE*) | **31.52%±1.38%** | 0.11±0.10 | 1.74± 0.06 |
| | AbRM (*CE&OF-II*) | 30.34%±2.71% | 0.23±0.22 | 1.63±0.19 |
| | SbRM(*CE*) | 30.54%±3.46% | 0.18±0.25 | 1.67±0.22 |
| | SbRM(*CE&OF-II*) | 30.76%±2.64% | **0.08±0.09** | 1.75±0.09 |

Table 3: Evaluating the clinical efficacy (survival rate) achieved by the proposed preference-based RL framework when hand-crafted and preference-based rewards are linear combined with different ratios.

| Ratios | AbRM (CE) | AbRM (CE&OF-I) | AbRM (CE&OF-II) | SbRM (CE) | SbRM (CE&OF-I) | SbRM (CE&OF-II) |
|---|---|---|---|---|---|---|
| 8:1 | 28.36%±3.09% | 28.07%±3.50% | 28.23%±3.82% | 29.10%±3.19% | 26.51%±2.47% | 28.94%±3.17% |
| 4:1 | 28.92%±2.82% | 27.41%±3.66% | 27.9%3±3.61% | 27.77%±3.16% | 28.45%±3.18% | 27.54%±3.09% |
| 2:1 | 26.30%±3.19% | 27.50%±3.64% | 27.71%±3.47% | 27.55%±3.40% | 27.81%±3.15% | 28.43%±3.12% |
| 1:1 | 27.42%±3.24% | 27.70%±3.73% | 27.62%±2.88% | 27.99%±2.9% | 28.49%±3.23% | 28.71%±3.50% |
| 1:2 | 26.96%±2.95% | 27.55%±2.90% | 28.66%±3.19% | 28.59%±2.94% | 28.79%±3.21% | 28.30%±3.29% |
| 1:4 | 27.46%±3.53% | 26.59%±3.40% | 28.25%±3.47% | 29.91%±1.93% | 28.75%±3.06% | 27.08%±3.64% |
| 1:8 | 27.15%±3.00% | 29.21%±2.42% | 27.67%±3.38% | 29.18%±2.45% | 28.55%±2.85% | 28.48%±3.28% |
| 0:1 (Ours) | **31.52%±1.38%** | **31.33%±1.18%** | **30.34%±2.71%** | **30.54%±3.46%** | **31.72%±1.08%** | **30.76%±2.64%** |

A.2    PREFERENCE-BASED REINFORCEMENT LEARNING ALGORITHMS

The preference-based Reinforcement Learning framework is composed of two main modules, *Preference-based Reward Learning* and *Preference-guided Agent Learning*. In *Preference-based Reward Learning*, the reward estimator parameterized by $\theta_P$ delivers step-wise rewards to the two agents parameterized by $\theta_A^1$ and $\theta_A^2$ based on their policy preference. In *Preference-guided Agent Learning*, the agents update their parameters so as to optimize the clinicians' objectives. The pair of policies performed by the two agents on the sampled subject is stored in the policy pool and leveraged for parameter update in reward estimator, with the aim to ensure higher expected return for the preferred policy. We list the pseudo codes for collaborative learning in Algorithm 1, *Preference-based Reward Learning* in Algorithm 2, and *Preference-guided Agent Learning* in Algorithm 3, respectively.

**Collaborative Learning**    Algorithm 1 illustrates the collaborative learning process between the two modules in order to estimate reward and learn policies in personalized treatment recommendation. In the beginning, the model parameters are randomly initialized (line 1), and the policy pools for the reward estimator and the two agents are created as empty sets (line 2). In each iteration, one subject is sampled from the training set for agent learning (line 3 to 5). At each simulation step, the two agents are asked to make decisions based on the current state and the reward estimator generates corresbonding step-wise reward for each of them (line 6 to 10). The subject's internal state keeps on updating until the simulation time has reached or the subject dies intermediately according to the underlying mathematical modeling. The policy pools of the two agents are augmented with the trajectories on the newest sampled subject (line 9 and 11), while the policy pool for the reward estimator is also updated (line 13) after computing the ground-truth preference label (line 12). After all the samples have been utilized for policy generation, the reward estimator minimizes the classification loss during policy preference inference with Algorithm 2 (line 16), while the RL agents optimize the expected return with Algorithm 3 (line 17).

---

**Algorithm 2** PREFERENCE-BASED REWARD LEARNING

---

**Require:**
   $\mathcal{D}_n$: sampled policy pairs in n-th iteration
   $\theta_P$: parameters to update in reward function
   $\gamma_P$: discounted factor on reward
   $\beta$: step size for parameter update
1: $L \leftarrow 0$
2: **for all** $(\tau^1, \tau^2, pre(\tau^1, \tau^2)) \in \mathcal{D}_n$ **do**
3:     $R(\tau^1; \theta_P) \leftarrow 0, R(\tau^2; \theta_P) \leftarrow 0$
4:     **for all** $(s_t^1, a_t^1, r_{\theta_P,t}^1, s_{t+1}^1) \in \tau^1$ **do**
5:         $R(\tau^1; \theta_P) \leftarrow R(\tau^1; \theta_P) + \gamma_P^t r_{\theta_P,t}^1$
6:     **end for**
7:     **for all** $(s_t^2, a_t^2, r_{\theta_P,t}^2, s_{t+1}^2) \in \tau^2$ **do**
8:         $R(\tau^2; \theta_P) \leftarrow R(\tau^2; \theta_P) + \gamma_P^t r_{\theta_P,t}^2$
9:     **end for**
10:     Compute $p(\tau^1 \succ \tau^2)$
11:     **if** $\tau^1 \succ \tau^2$ **then**
12:         $L \leftarrow L + \log p(\tau^1 \succ \tau^2)$
13:     **else if** $\tau^2 \succ \tau^1$ **then**
14:         $L \leftarrow L + \log \left(1 - p(\tau^1 \succ \tau^2)\right)$
15:     **else if** $\tau^1 \sim \tau^2$ **then**
16:         $L \leftarrow L + 0.5 \log p(\tau^1 \succ \tau^2) + 0.5 \log \left(1 - p(\tau^1 \succ \tau^2)\right)$
17:     **end if**
18: **end for**
19: Update $\theta_P \leftarrow \theta_P - \beta \Delta_{\theta_P} L$
20: **return** $\theta_P$

---

**Preference-based Reward Learning**    Given pairs of policies with corresponding preferences, the reward estimator updates its parameters to maximize the probability that the preferred policy achieves

higher expected return than the other. As shown in Algorithm 2, the discounted expected returns achieved by each agent are firstly calculated respectively for each sampled policy pair (line 3 to 9). Then the probability that policy $\tau^1$ is preferred to $\tau^2$ is positively correlated to the expected return of $\tau^1$, and is computed as (Agresti & Kateri, 2011) introduced (line 10). Hence $p(\tau^2 \succ \tau^1)$ is equal to $1 - p(\tau^1 \succ \tau^2)$. Then the loss value is computed considering different kinds of preference relationships between the two policies (line 11 to 17). Incomparable policy pairs are also leveraged in reward learning for better preference space exploration (line 15 to 16).

---

**Algorithm 3** PREFERENCE-GUIDED AGENT LEARNING

---

**Require:**
    $\Gamma_n$: sampled policies from one agent in n-th iteration
    $\alpha$: step size for parameter update
    $\mathcal{M}$: one of the two reward assignment methods
    $L_{\theta_A}$: loss function in any deep RL approach parameterized by agent parameters $\theta_A$
1:  $\varepsilon = \emptyset$
2:  **for all** $(s_t, a_t, r_{\theta_P, t}, s_{t+1}) \in \Gamma_n$ **do**
3:     **if** $\mathcal{M}$ is *Action-based Reward Modification* **then**
4:        $r_t \leftarrow r_{\theta_P}(s_t, a_t)$
5:     **else if** $\mathcal{M}$ is *State-based Reward Modification* **then**
6:        $r_t \leftarrow h_{\theta_P}(s_t) - h_{\theta_P}(s_{t-1})$
7:     **end if**
8:     $\varepsilon \leftarrow \varepsilon \cup \{(s_t, a_t, r_t, s_{t+1})\}$
9:  **end for**
10: Update $\theta_A \leftarrow \theta_A - \alpha \Delta_{\theta_A} \sum_{x \in \varepsilon} L_{\theta_A}(x)$
11: **return** $\theta_A$

---

**Preference-guided Agent Learning**   Each agent updates their parameters individually as Algorithm 3 depicts. The agent receives rewards computed by either *Action-based Reward Modification* (line 3 to 4) or *State-based Reward Modification* (line 5 to 6). Then we leverage $(s_t, a_t, r_t, s_{t+1})$ to update the agent model implemented by any deep Reinforcement Learning approach.

## A.3   SIMULATION PLATFORM DESIGN

### A.3.1   MEDICATION RECOMMENDATION FOR GENERAL CANCER

**Survival Analysis**   Within time interval $(t - 1, t]$, where $(1 \leq t \leq 6)$, the survival status is assumed to depend on both the current tumor size $y_t$ and the toxicity level $x_t$. The probability of a patient's death is modeled as follows:

$$\text{Hazard function: } \lambda(t) = \exp(-4 + y_t + x_t), \quad \text{Cumulative hazard function: } \Delta\Delta(t) = \int_{t-1}^{t} \lambda(s)d(s),$$

$$\text{Survival function: } \Delta F(t) = \exp(-\Delta\Delta(t)), \quad \text{Death probability: } p_{\text{death}} = 1 - \Delta F(t).$$

**Implementation Details**   The action space is discrete and the dosage amount decisions are selected among 4 options: 0.1, 0.4, 0.7, 1.0 (Fürnkranz et al., 2012). For state initialization, the tumor size and the toxicity level in the $0^{th}$ month are generated independently from the uniform distribution $\mathcal{U}(0, 2)$. The simulation terminates after $t = 6^{th}$ month or if the patient dies intermediately.

**Model Implementation and Training**   For 6-month simulation, we randomly sample $10,000$ subjects for training, $2,000$ for validation, and $2,000$ for testing. The neural networks for all deep learning approaches including *preference learning* and *reinforcement learning* share the similar network structure and hyper-parameters: 2 fully-connected layers, the first followed by ReLU activation and the second followed by different activation functions for different approaches. In one epoch, the agent gets updated after seeing all the training samples. The learning rate is set to $0.01$ and all the networks converge after 400 epochs. For deep RL methods, we set the discount factor $\gamma$ to 1.

A.3.2 BLOOD PURIFICATION RECOMMENDATION FOR SEPSIS

**Mathematical Modeling in Simulation**    Sepsis is initiated by spillover of pathogens into blood, where the pathogen is allowed to spread throughout the organism in which systemic inflammation takes place (Stojkovic et al., 2016). Motivated by the promising results of blood purification in other critical illness conditions like acute kidney failure (Ronco et al., 2000), blood purification has gained attention as a potentially effective solution for septic subjects (Rimmelé & Kellum, 2011). In blood purification treatment, the patient is connected to an extracorporeal hemoadsorption device that removes harmful particles from the blood and leads the patient towards a healthy state.

We employ the mathematical model derived by Song *et al.* to simulate the acute inflammation process in response to an infection (Song et al., 2012). Both heuristic knowledge about the mechanism underlying infection and real measurements from experiments on CLP-induced septic rats were leveraged for the model design. The distribution of initial physiological features and their interactions are derived from domain knowledge. The initial physiological features that characterize a subject accords with the probability distributions based on real experimental measurements for septic rats. The parameters in transition functions are calibrated so that the generated trajectories closely follow experimentally observed temporal patterns in septic rats.

Figure 12 demonstrates the feature interaction network. There are 19 physiological features that govern sepsis dynamics, 8 of which are observable (features above the horizontal dashed line) while the remaining 11 are conceptual variables (features below the horizontal dashed line). When a blood purification operation is made, three components in the circulation are eliminated (features marked by red dashed ring), i.e., activated neutrophils $N_a$ and the pro- and anti-inflammatory mediators $PI$ and $AI$. Besides effects from the blood purification operation, the variables influence each others' progression through Ordinary differential equations (ODEs).

**State Transition**    There are 18 ODEs to describe feature interactions and 3 ODEs for the hypothetic mechanism of blood purification. The hypothetic mechanisms of action of the blood purification are implemented by assuming the hemoadsorption device eliminates only three components in the circulation: activated neutrophils ($N_a$), pro-inflammatory mediators ($PI$), and anti-inflammatory mediators ($AI$) during the treatment period. We here only show the transition equation of these three key features with and without operation, ODEs concerning other features can be found in (Song et al., 2012).

The variable **PI** stands for the extent of the systemic inflammation and progresses as follows:

$$\frac{dPI}{dt} = \Big( \frac{B/B_\infty}{h_{PI\_B} + B/B_\infty} \Big( 1 - \frac{D^n}{h_{PI\_D}^n + D^n} \Big) \Big( 1 - \frac{AI^n(1 - PI)}{h_{PI\_AI}^n + AI^n} \Big) \tag{3}$$

$$+ \Big( 1 - \frac{B/B_\infty}{h_{PI\_B} + B/B_\infty} \Big) \frac{D^n}{h_{PI\_D}^n + D^n} \Big( 1 - \frac{AI^n(1 - PI)}{h_{PI\_AI}^n + AI^n} \Big)$$

$$+ \frac{B/B_\infty}{h_{PI\_B} + B/B_\infty} \frac{D^n}{h_{PI\_D}^n + D^n} \Big( 1 - \frac{AI^n(1 - PI)}{h_{PI\_AI}^n + AI^n} \Big) - PI \Big) \frac{1}{\tau_{PI}},$$

$$PI(t'+1) = \begin{cases} PI(t') + \frac{dPI}{dt}(t') & \text{If no operation is performed} \\ PI(t') + \frac{dPI}{dt}(t') - \frac{PI}{h_{PIHA} + PI} & \text{Otherwise} \end{cases}, \tag{4}$$

where $B$ is the population of bacteria in the peritoneum, $D$ is a coarse-grained representation of integrated tissue damage, variables $h_{PI\_B}, h_{PI\_D}, h_{PI\_AI}, \tau_{PI}$ are subject-specific parameters, $B_\infty$ is a predefined upper bound of $B$, $h_{PIHA} = 0.3$ and $n = 3$.

Table 4: Configurations for Sepsis treatment simulation. $h$ represents hour in the simulation platform.

| Step Size $\tau$ | Horizon Length $T$ | Operation Time Interval $L$ | Decision-making Frequency $f$ | Duration per Operation $l$ |
|---|---|---|---|---|
| $\tau = 0.1$ h | $T = 100$ h | $5^{th}$ to $18^{th}$ h | $f = 2$ h or 4h | $l = 2$ h or 4h |

The variable **AI** describes the level of the anti-inflammation corresponding to systemically acting anti-inflammatory mediators and gets updated as follows:

$$\frac{dAI}{dt} = \Big( \frac{PI^{n_1}}{h_{AI\_PI}^{n_1} + PI^{n_1}} \Big( 1 - \frac{N_a/N_\infty}{h_{AI\_N_a} + N_a/N_\infty} \Big) + \Big( 1 - \frac{PI^{n_1}}{h_{AI\_PI}^{n_1} + PI^{n_1}} \Big) \frac{N_a/N_\infty}{h_{AI\_N_a} + N_a/N_\infty}$$

$$+ \frac{PI^{n_2}}{h_{AI\_PI}^{n_2} + PI^{n_2}} \frac{N_a/N_\infty}{h_{AI\_N_a} + N_a/N_\infty} - AI \Big) \frac{1}{\tau_{AI}}, \tag{5}$$

$$AI(t'+1) = \begin{cases} AI(t') + \frac{dAI}{dt}(t') & \text{If no operation is performed} \\ AI(t') + \frac{dAI}{dt}(t') - \frac{AI}{h_{AIHA}+AI} & \text{Otherwise} \end{cases}, \tag{6}$$

where variables $h_{AI\_PI}, h_{AI\_N_a}, \tau_{AI}$ are subject-specific parameters, $N_\infty$ is a predefined upper bound of neutrophils, $h_{AIHA} = 0.3$, $n_1 = 1$ and $n_2 = 3$.

The variable **$N_a$** represents the activated blood neutrophils and transits in each simulation step as follows:

$$\frac{dN_a}{dt} = \underbrace{\frac{N_r PI^n}{h_{N_r\_N_a}^n + PI^n} \frac{1}{\tau_{N_r\_N_a}}}_{\text{transmission from } N_r \text{ to } N_a} + \underbrace{\frac{N_p PI^n}{h_{N_p\_N_a}^n + PI^n} \frac{1}{\tau_{N_p\_N_a}}}_{\text{transmission from } N_p \text{ to } N_a} - \frac{N_a}{\tau_{N_a}} - \underbrace{\frac{N_a PI^n}{h_{N_a\_N_s}^n + PI^n} \frac{1}{\tau_{N_a\_N_s}}}_{\text{transmission from } N_a \text{ to } N_s}, \tag{7}$$

$$N_a(t'+1) = \begin{cases} N_a(t') + \frac{dN_a}{dt}(t') & \text{If no operation is performed} \\ N_a(t') + \frac{dN_a}{dt}(t') - \frac{N_a/N_\infty}{h_{N_a HA}+N_a/N_\infty}(t') & \text{Otherwise} \end{cases}, \tag{8}$$

where $N_r$ is resting blood neutrophils, $N_p$ is blood neutrophils, $N_s$ is neutrophils sequestered in the lung capillaries, variables $h_{N_r\_N_a}, h_{N_p\_N_a}, h_{N_a\_N_s}, \tau_{N_r\_N_a}, \tau_{N_p\_N_a}, \tau_{N_a\_N_s}$ are subject-specific parameters, $h_{N_a HA} = 0.3$, and $n = 3$.

**Survival Analysis**    The survival status of the subject only depends on the value of the systemic pro-inflammatory response $PI$ at the end of the simulation. When the $PI$ value at the last time-step is smaller than the pre-defined threshold $0.5$, then the subject is assumed to be alive, otherwise dead. Note that after the blood purification process, the $PI$ value reduces as time passes, hence one cannot conclude whether the subject is alive in the intermediate time-steps. After the pre-defined simulation horizon is reached, we can confirm which subjects survive with the help of treatment. The mathematical model is quite different from the general Cancer Treatment model where subjects have a probability to die intermediately.

**Implementation Details**    Due to phenotype differences, some subjects survive without any blood purification operation while some die. This is consistent with laboratory experiments where $30\%$ of rats survived till seven days while the remaining died between two to five days after CLP (Zhao et al., 2009). We call the survivor group *Survival Population* and the non-survivor group *Death Population*. The survival status of the *Survival Population* gets no influence from blood purification operations. Subjects from *Death Population* have the potentials to survive if proper treatment policies are delivered. Since we are primarily concerned about the outcomes on subjects from *Death Population*, we only sample subjects from the *Death Population* in this paper to train and evaluate treatment policies.

There are a few hyper-parameters that should be set in advance: 1) Simulation step size $\tau$: every $\tau$ time, the simulator updates the internal status of subjects by computing the ODEs with feature values from the last simulation step and the current action. 2) Simulation horizon length $T$: we can evaluate the performance of a policy by checking outcomes of subjects after time $T$. 3) Valid time

range $L$ for patients to receive treatment: operations can take place at any time-step ($L = [0, T - 1]$) or be constrained to predefined time intervals ($L \subsetneq [0, T - 1]$). 4) Frequency of decision-making $f$ : subjects can receive operations at each simulation step $\tau$ or less frequently. 5) Duration of each blood purification operation $l$: it takes some costs to turn on/off the purification device and it is also unrealistic to attach and detach the device from the subject too frequently. Therefore, there should be a pre-defined value for the purification duration to rule out the possibility of too frequent actions.

To generate testable hypotheses that guide future laboratory experiments (Song et al., 2012; Stojkovic et al., 2016), the simulation of sepsis evolution should be configured to make the generated trajectory closely follow experimentally observed temporal patterns (Song et al., 2012). Further, several constraints can be imposed on the simulation in accordance with previous blood purification studies (Song et al., 2012; Stojkovic et al., 2016). Therefore we use the configuration listed in Table. 4 for experiments.

**Model Implementation and Training**  We randomly sample 3,000 subjects for training, 1,000 for validation, and 1,000 for testing. Implementation details of the deep RL approaches are similar to those mentioned in the Cancer task, except that the backend network is LSTM-based since this is a POMDP.

Learning efficient treatment policies for Septic subjects is more difficult for Cancer due to the larger state space and the partially observable environment. Therefore, we adopt the following methods to ensure robust learning: 1) Mini-batch gradient descent with batch size 10,000 is adopted to update parameters in reward estimator and RL agents. 2) The learning rate for RL agents is $0.01$ while $0.001$ for the reward estimator. 3) As discussed in Experiment Section, experience replay makes the estimated reward positively proportional to the *Survival Rate*. We randomly extract policy pairs from the latest 30,000 samples for model updates.

Figure 11: Distribution of state features (tumor size and toxicity level) from Cancer subjects without tre

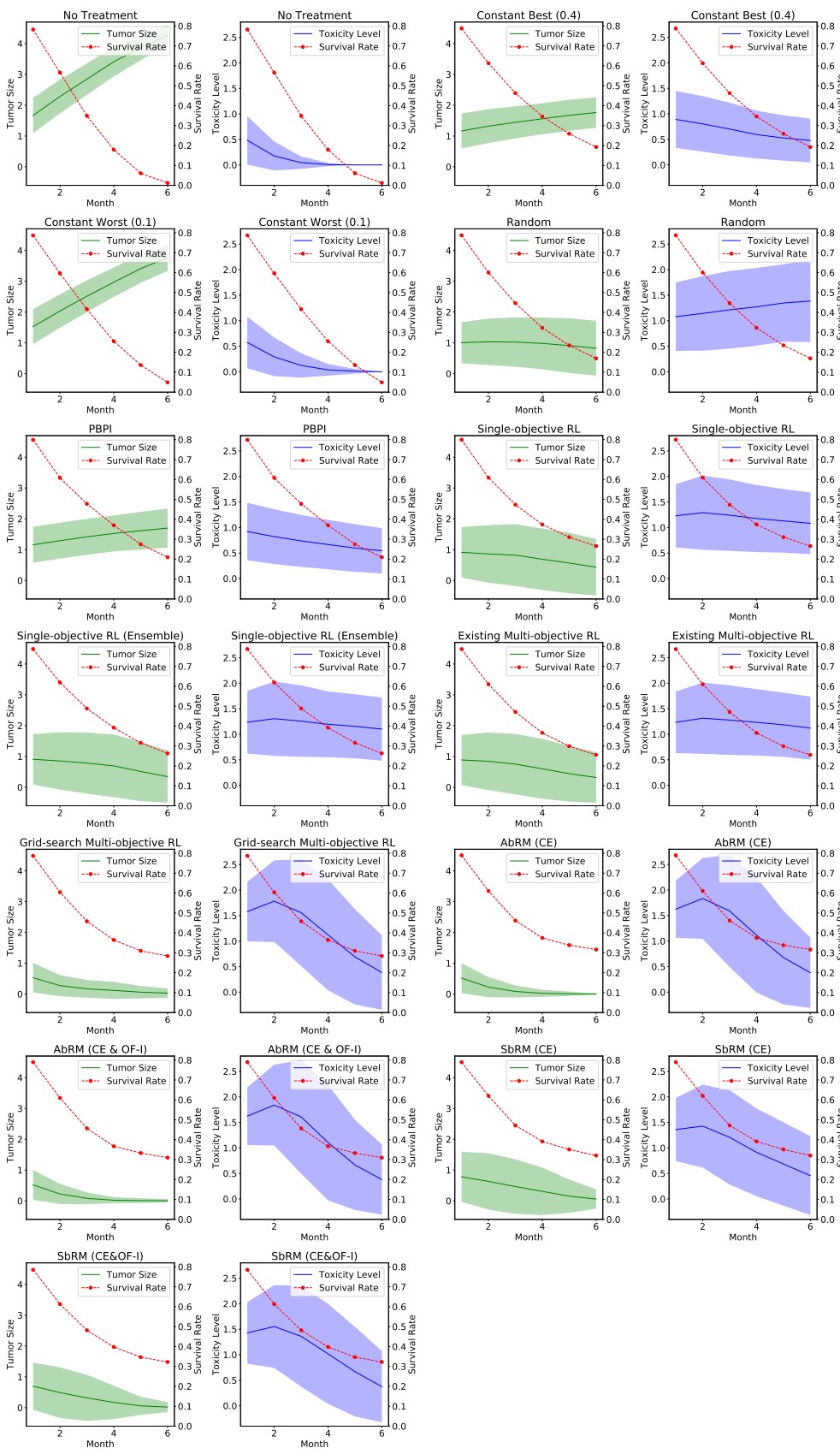

Figure 12: Interaction network of inflammatory responses and hypothetic hemoadsorption mechanisms of action in CLP-induced sepsis. Nodes in green represent components in peritoneum, nodes in orange stand for blood components, and nodes in purple stand for lung components. Edges represent network interactions under blood purification treatment compiled from literature. When blood purification is performed, only features $PI$, $AI$ and $Na$ are influenced (marked by red dashed rings).

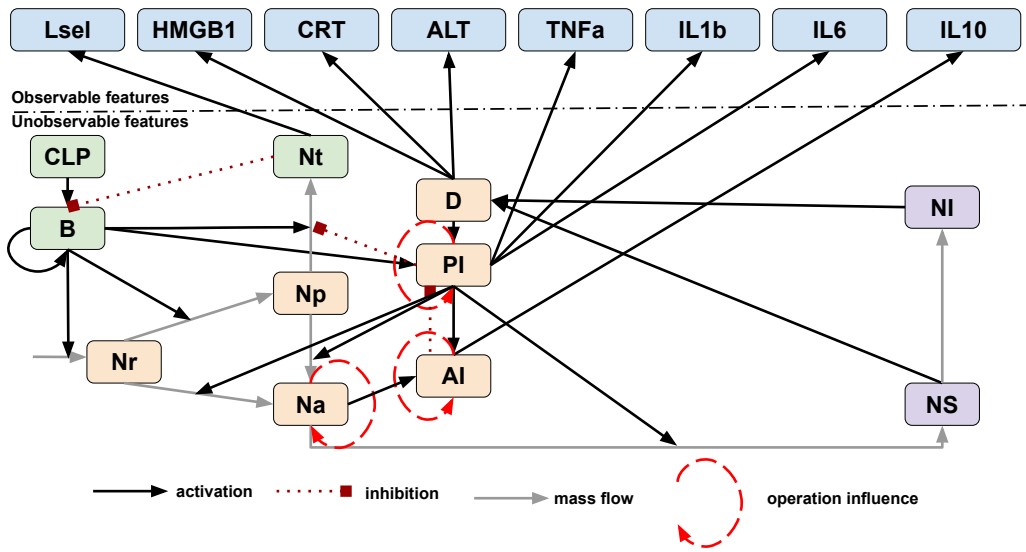

Figure 13: Distribution of 19 state features from Septic subjects without treatment or with treatment from SbRM.

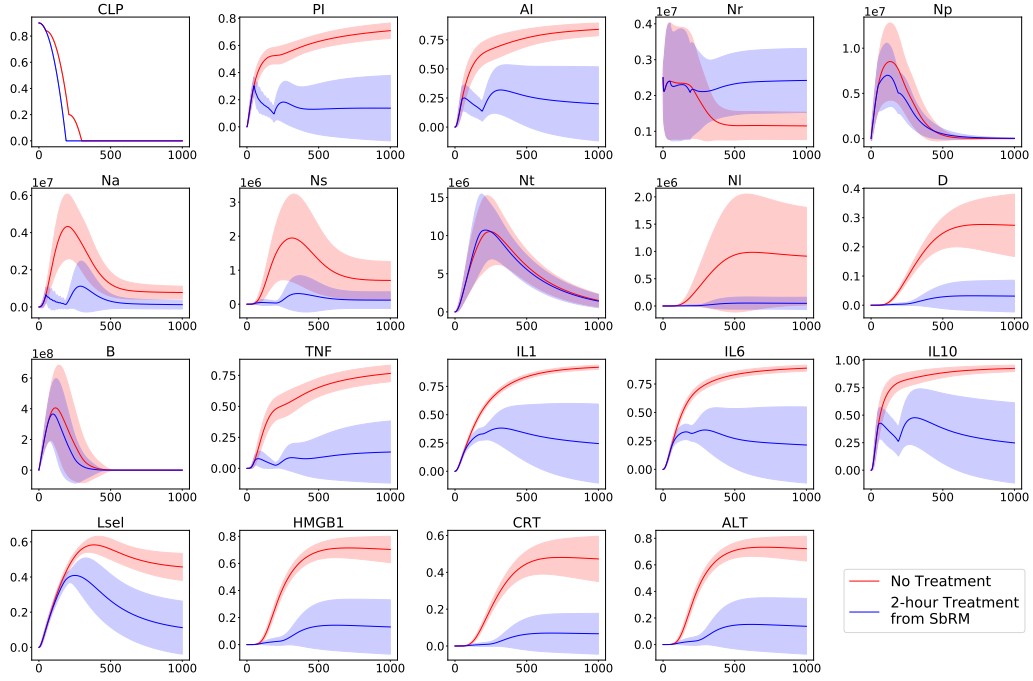

