# OpenReview forum: "An Examination of Preference-based Reinforcement Learning for Treatment Recommendation"
_ICLR.cc/2021/Conference — Reject_

### Official Review · AnonReviewer1 · 2020-10-25
**Lacks comparison with sota**

**Rating:** 4
**Confidence:** 4

**Review:**

This paper describes a preference based reinforcement learning framework for treatment recommendation tasks. The authors use a model-based environment to simulate the progression of disease and propose a preference based approach. They claim they can use this framework with any off-the-shelf RL algorithm to generate better survival results. The experiments are carried out with policy gradient. The authors also compared several different ways to design reward functions, and propose to use incomparable samples with a 0.5/0.5 distribution in the Bradley-Terry model, instead of discarding such samples, to improve the performance. The paper is generally well-written and easy-to-follow. Despite the technical contributions listed above, I would suggest rejection due to the following concerns.

1. Innovation
The paper lacks citations and comparisons with several recent works in this field, such as [r1, r2]. The approach proposed is similar and the innovation seems marginal. I suggest the authors include a comparison of methodology and, if possible, experiments comparing their approach with state of the art preference based RL algorithms in the revision.

2. Lack of implementation details
There are a few details I didn't find reading the manuscript, for example, how to query human responses on line 12 of Algorithm 1, what is the guideline for human experts to rate different treatment plans, implementation details and hyper-parameters used in the agent learning algorithm. The lack of such details makes it difficult for other researchers to reproduce the results shown in the paper.


[r1] Christiano, Paul F., et al. "Deep reinforcement learning from human preferences." Advances in Neural Information Processing Systems. 2017.
[r2] Ibarz, Borja, et al. "Reward learning from human preferences and demonstrations in Atari." Advances in Neural Information Processing Systems. 2018.

---

> ### Author Response · Authors · 2020-11-23
> **Related preference learning work comparison**
>
> We have cited Christiano’s work but missed Ibarz’s work in our paper. We didn’t compare explicitly with the existing approach for learning from preferences since all of the preference-based reward estimators (including ours) follow the Bradley-Terry model (Bradley and Terry, 1952) for estimating score functions from pairwise preferences, and is the specialization of the Luce-Shephard choice rule (Luce, 2005; Shepard, 1957) to preferences over trajectory segments. Some latest work concerning learning reward functions based on ranking [1, 2] also adopted the same loss function for reward learning. The main differences of our work from Christiano’s and Ibarz’s work are two-folds: 1) we do not need explicit human involvement to annotate preference over policies during model training since the preferences are generated automatically based on some pre-defined metric, e.g., maximizing the survival rate, reducing the operation times for Sepsis experiments; 2) the majority of existing work disregarded incomparable pairs while we treat incomparable policies equally. As discussed in Section 5.4 and illustrations from Fig. 3, we find that incomparable policies are efficiently utilized for better preference space exploration (preference approaching 0.5 as expected in Fig. 3a), more samples for preference model update (all the 10,000 samples from the training set participate in the loss function minimization in Fig. 3b), and much higher clinical efficacy (more than 30% survival rate achieved after the model converges in Fig. 3c).
>
> [1] Brown, Daniel S., et al. "Extrapolating beyond suboptimal demonstrations via inverse reinforcement learning from observations." arXiv preprint arXiv:1904.06387 (2019).
>
> [2] Brown, Daniel S., Wonjoon Goo, and Scott Niekum. "Better-than-demonstrator imitation learning via automatically-ranked demonstrations." Conference on Robot Learning. 2020.

---

> ### Author Response · Authors · 2020-11-23
> **Implementation details**
>
> Overall, two policies are compared firstly based on clinical efficacy, i.e., capability to save subject’s life; if both two lead to survivals, then policy influences on subject’s health conditions are considered for a further comparison. If no policy is preferred to the other after comparison, then the two are incomparable.
>
> In section 5.3, we briefly describe the preferences to consider for Cancer and Sepsis treatment. We elaborate here how different evaluation metrics of preferences are utilized to automatically compare two policies (by EVALUATEPREFERENCE function in Algorithm 1) as follows:
>
> 1) Cancer:
>
> - CE: Maximizing survival rate (results in main/Table 1). The policy saving the subject’s life is preferred to the policy leading to subject’s death. Otherwise, the two policies are incomparable.
>
> - CE&OF-I: Maximizing survival rate and mitigating negative impacts in terms of minimizing sum of tumor size and toxicity level at the end of simulation (results in main/Table 1). The policy saving the subject’s life is preferred to the policy leading to subject’s death. If both lead to subject’s survival, then the policy with a smaller sum value of tumor size and toxicity level is preferred. Otherwise, the two policies are incomparable.
>
> - CE&OF-II: Maximizing survival rate and mitigating negative impacts in terms of minimizing the highest toxicity level and final tumor size during simulation (results in appendix/Table 2). The policy saving the subject’s life is preferred to the policy leading to subject’s death. If both lead to subject’s survival, then the policy with a smaller highest toxicity level and a smaller final tumor size simultaneously is preferred. Otherwise, the two policies are incomparable.
>
> 2) Sepsis
>
> - Optimizing clinical efficacy only: Maximizing survival rate (results in main/Fig.1(a)). The policy saving the subject’s life is preferred to the policy leading to subject’s death. Otherwise, the two policies are incomparable.
>
> - Optimizing clinical efficacy & mitigating negative impacts: Maximizing survival rate and reducing number of operations (results in main/Fig.1(b)). The policy saving the subject’s life is preferred to the policy leading to subject’s death.  If both lead to subject’s survival, then the policy with fewer times of operations is preferred. Otherwise, the two policies are incomparable.

---

### Official Review · AnonReviewer4 · 2020-10-25
**When reinforcement learning is considered for treatment recommendation, there is a trade-off between survival rate and other factors and side effects. In this paper, the authors use preferences to learn the reward that will guide the RL agents towards behaviour that matches the objectives that are used to assess agents' performance.**

**Rating:** 4
**Confidence:** 3

**Review:**


This is a relatively comprehensive evaluation. The authors know the related literature, and a number of experiments are presented to show different aspects of the methods. The work is promising. Writing is great, and the authors can write beautiful sentences, but the overall structure has some flaws.

I am not happy with the way the authors split their manuscript into the main paper and the appendix. The current version of the main paper is not self-contained, and it does not provide minimum explanation of the methods. I was very perplexed after reading the main part before I read the appendix because the core algorithm is not explained in the main part of the paper. It was hard to follow the experiments in sec. 5, when one is unsure how the methods work. The experiments section itself makes frequent jumps to the appendix. This content would be much more appropriate for a journal paper, in which most of the appendix would be nicely integrated with the main body.

It is not entirely clear what is the new contribution in this paper. When algorithm 1 is introduced in section 4.2, the authors do not say which components are new or how they extend the existing literature.

The EVALUATEPREFERENCE procedure in algorithm 1 is not defined, yet it is a very important component. The pseudocode of algorithm 1 indicates that entire trajectories are compared to elicit preferences, but this may lead to suboptimal behaviour. Note that if two policies are suboptimal, they may make suboptimal decisions in different states, so preferences at the level of individual states would be more appropriate. This important component of the algorithm is unclear to me in the paper.

Following on the previous question, I should ask what the human expert would need to do if this algorithm was applied in practice. Specifically, what would be displayed to the human expert, and what the human expert would need to do or what they would need to compare and judge? The simulator (which would be the real environment) and the algorithms are tightly entangled in the appendix, and for this reason it is hard to imagine how this solution would work with real data and real human input.

Since the authors' goal is to show that their method provides better rewards than the handcrafted rewards known in the existing literature, perhaps some discussion of those rewards could be added. How were those rewards derived? Did the human designers have the same objectives in mind? Could one derive a better handcrafted reward for these experiments? Sufficient evidence should be provided that the handcrafted rewards used in comparisons are not trivial. I am saying this because the preference-based reward is derived from preferences that match the quantities that constitute the objective of learning, so it is not that surprising that preference-based reward leads to better performance according to the corresponding objectives.

Small issues:

In sec. 1, the authors mention open access large-scale Electronic Health Records. Examples of such publicly available records should be provided.

The authors should clarify what they mean in this sentence "the linearly weighted reward function induces negative interface between objectives".

Several symbols are not defined in section 2.1, e.g., a1, a2, m1, m2.

Formatting on p. 3 is incorrect.

What are "hill" equations on p. 3?

It would be good if the authors could explain equations that are in lines 5 and 8 in Algorithm 2. What would be an intuitive explanation of L in this algorithm?



This paper addresses a very important problem, and the results could be significant. I think that the future readers would be confused if this paper was accepted in this form. It would be sensible to turn it into a longer journal paper and clarify the key components.

---

> ### Author Response · Authors · 2020-11-23
> **Answers to main questions**
>
> 1. EVALUATEPREFERENCE procedure
>
> Once we have determined the metric/rule that determines one policy is preferred to the other policy, then we can conduct EVALUATEPREFERENCE procedure automatically. Take the CE&OF-I case mentioned in Section 5.1 for Cancer treatment recommendation as an example. The goal of CE&OF-I is to maximize survival rate and mitigate negative impacts in terms of minimizing sum of tumor size and toxicity level at the end of simulation (results in main/Table 1). The policy saving the subject’s life is preferred to the policy leading to the subject's death. If both lead to the subject's survival, then the policy with a smaller sum value of tumor size and toxicity level is preferred. Otherwise, the two policies are incomparable.
>
> 2. Preference at individual states or at whole policy
>
> We consider preference over policy rather than a single state for two reasons. Firstly, it is very difficult to conclude that one action is preferred to the other given a state in treatment recommendation, either in pre-defined metric way or consulting an expert.  That’s also the reason when hand-crafted rewards are utilized for policy learning, intermediate steps are normally assigned with 0 reward and the final step is assigned with positive reward for positive outcome or negative reward for negative outcome. Secondly, we cannot compare two random states or state-action pairs directly since they may come from different subjects with different health conditions. For two policies starting with the same initial state, we can have a reasonable comparison by checking the outcomes of different treatment strategies after observing the whole policies on the same subject.
>
> 3. How to work with real data and real human input
>
> State transition parameters in Sepsis experiments characterize each subject and are sampled using Markov-Chain Monte Carlo (MCMC) sampling of their posterior parameter distributions. The parameters for each subject are accepted if they are compatible with their respective distributions and the observations in the resulted simulation are close to collected experimental data from 23 rats. Suppose the simulator can emulate quite well the acute inflammation process in response to an infection, then human input is the metric to evaluate preference over two policies, such as the survival rate, the number of operations, etc.
>
> 4. Comparison with hand-crafted reward
>
> To have a fair comparison between preference-based reward and hand-crafter reward, we show performance of Grid-search Multi-objective RL for both Cancer and Sepsis (Table 1, Figure 6 and Figure 1), and grid-reward design for positive and negative outcomes for Cancer (Figure 5).
>
> In Fig. 5 (Appendix), we first show the sensitivity of policy behaviors to small changes in handcrafted rewards leads to unstable clinical efficacy even when the relative importance of obtaining positive outcomes against negative ones keeps unchanged. The three heatmaps of the agent’s performance in response to different reward scalars (121 combinations) reflect the difficulty in specifying an appropriate reward function to enable policy learning with the optimal clinical efficacy during treatment recommendation.
>
> In Fig. 6 (Appendix), we further show the difficulty of selecting reward scalars for three factors – survival rate, last tumor size and maximum toxicity level – in the grid-search Multi-objective RL approach to appropriately prioritize the clinical efficacy over negative impacts. We can observe that policy learning from preference-based reward can have the highest survival rate compared with policies learning from all kinds of linear weighted sum of the three factors.

---

> > ### Comment · AnonReviewer4 · 2020-11-24
> > **read answers to main questions**
> >
> > Thank you for your comprehensive reply. As I said in my review, you are doing solid work, but the original draft was deficient. I appreciate your effort, and many thanks for writing up the answers. I don't have any other questions to you. We will further discuss your submission with the other reviewers.
> > All the best.

---

> ### Author Response · Authors · 2020-11-23
> **Answers to small issues**
>
> - Linearly weighted reward function induces negative interference between objectives: when multiple objectives are included in one optimization function, then it could happen when one objective is optimized while the other isn’t.
>
> - Some notations: $a_1, a_2, b_1, b_2, m_1, m_2$ are hyper-parameters designed in the mathematical model proposed by Zhao et al. (2009). We adopt their defaulted values in our experiments.
>
> - Hill equations from state transitions in Section 3.2: Several molecular interactions in cellular systems exhibit sigmoidal response curve to variations in the input concentrations. Such a response curve is typically represented by Hill equation: $Y=I^{nH}/(K_{0.5}^{nH}+I^{nH})$, where $Y$ is the output response and $I$ is the input concentration. Hill equation involves two parameters, Hill Coefficient ($nH$) and half-saturation constant ($K_{0.5}$). While Hill coefficient characterizes the sensitivity of the response, the half-saturation constant quantifies the threshold concentration required for 50% output response.
>
> - Equations in line 5 and 8 from Algorithm 2: expected return computation, i.e., sum of discounted reward
>
> - Loss in Algorithm 2: as discussed in Section 5.4, there are cases when two policies are incomparable. Since no clear preference conclusion can be drawn between the two incomparable policies, the majority of existing work in preference learning disregarded them directly. After the performance analysis shown in Fig.3, we find that excluding incomparable pairs from the training set leaves the parameterized model exploring the preference space arbitrarily and inferring random preference over two policies although they are incomparable. To avoid arbitrary exploration in the preference space, we handle the incomparable pairs with a simple approach: treating both policies from the incomparable pair equally, i.e., $I(\pi_m(s_i) ≻ \pi_n(s_i)) = I(\pi_n(s_i) ≻ \pi_m(s_i)) = 0.5$. With the small but important augmentation to the preference indicator function, incomparable policies are efficiently utilized for better preference space exploration (preference approaching 0.5 as expected in Fig. 3a), more samples for preference model update (all the 10,000 samples from the training set participate in the loss function minimization in Fig. 3b), and much higher clinical efficacy (more than 30% survival rate achieved after the model converges in Fig. 3c).

---

> > ### Comment · AnonReviewer4 · 2020-11-24
> > **read answers to small issues**
> >
> > Thank you for taking the time to write these explanations. The paper would read much better if these details were in it. I hope that you will add them.

---

### Official Review · AnonReviewer3 · 2020-10-28
**Interesting concepts and extensive experimental justification; Unfortunately incomplete and unclear technical development**

**Rating:** 4
**Confidence:** 4

**Review:**

#### **Summary**
This paper develops a preference-based framework for Reinforcement Learning in Healthcare, focusing on treatment recommendation. The paper highlights several factors where policies trained with preferences may be better suited for use in healthcare applications and extensively evaluates these factors using two simulated care scenarios.


#### **Assessment**
The foundational concepts underlying this paper are strong and the authors make good points framing the potential advantage of using preference-based rewards over what is termed a “handcrafted reward”. Reward design is an open challenge in Reinforcement Learning, particularly in Healthcare settings, and this paper proposes an intriguing way to perhaps avoid explicitly designing a reward while still providing improved performance. However there are significant gaps in the technical development of the paper that greatly reduce the clarity and perceived significance of the presented work (several examples highlighted in the “Weaknesses” section below). Several claims are made in the introduction and set-up of the experiments that are not fully supported, discussed or justified. Finally, there are some major concerns about the suitability of the simulated environments to adequately evaluate the contributions of the proposed methods (more in the “Weaknesses” section below) in light of the applicability to real world dynamics.


#### **Strengths**
- The paper presents an appealing rationale for the use of preference-based RL in complex sequential decision making problems where reward design is a significant challenge. The paper is firmly grounded in relevant literature. There are several papers that I expected to be included (see Relevant Literature in the “General Comments” section) but their omission is not a critical error.
- The experimental investigation is ambitious and covers most, if not all, relevant questions regarding the utility of preference-based rewards in treatment recommendation. Several relevant baselines are used to compare and contrast the two proposed preference based RL approaches, highlighting the apparent strengths of the proposed method.
- Specifically, I found the analysis and experiments about transferability and how the proposed approach handles “incomparable policies” to be really interesting. Only transferring the parameters of the estimated reward functions instead of the policies themselves is a really interesting idea and could stand a far more detailed and extensive study in its own right.
- The introduction of additional simulated treatment domains will be a nice contribution to the community as there are only a small handful of stable simulators available within the RL for healthcare domain.


#### **Weaknesses**

I found several weaknesses in the presentation and development of the proposed approach. Throughout the paper concepts are not fully detailed and are only vaguely referred to, seemingly in expectation that the reader implicitly understands or knows what is meant. The major offense in this direction is that the “human” preference-based rewards are never explicitly defined. It’s not clear from the formulation how the returns or observations from the environments are used to compute the rewards under this framing. I read the paper and appendices several times to look for a formalization of these reward definitions and was disappointed to not find them. The closest I could come was in the clinical efficacy and “other factors” metrics. But I didn’t feel that these could be the defined preferences because the authors continually draw a distinction between hand designed rewards and preference-based rewards determined by human intentions.

Along this point, there are continual references to human preferences or human feedback guiding reward design. However, isn’t this just a human-in-the-loop version of handcrafting a reward? Also, it’s unclear that a clinician will be able to interpret the neural network policies and learned reward representations to provide adequate guidance and preference determination. In the introduction the following statement is made: “Fortunately, qualitative feedback according to human’s preferences can be easily obtained and efficiently leveraged…” This is unfortunately not true in regards to clinical decision making without resulting in overly biased evaluations. Individual clinicians differ in their interpretation of patient conditions as well as the appropriate route of treatment. Without belaboring the point, it’s not clear how establishing the preferences between policies is not just a different form of handcrafting a reward function. Missing from the paper is a discussion about how these preferences would be obtained and integrated into the development of learning a policy. Based on the conclusion it appears that this wasn’t actually incorporated in this paper leaving major questions about the proposed direction and claims made in the paper about handcrafted vs. preference based rewards. Pessimistically, the observed gains with AbRM and SbRM could be attributed to ensembling (not altogether novel given recent advances in offline RL utilizing multiple agents to deal with overestimation within the learned value functions) or more informative reward design.

On this point regarding overestimation. There are major concerns that the preference based reward and how the individual reward functions (and agents) are trained will largely suffer from overconfidence, a standard problem in RL. Simulators largely cover over this limitation because unrealistic, non-physical behaviors or actions are still supported. In practice, RL in healthcare will be used in offline, off-policy settings where overconfidence and extrapolation to actions not seen in the training set will lead to ineffective and, in the worst case, fatal treatment policies (see Gottesman, et al [2019]). Simulators can only get us so far without specific guarantees about how realistic and representative they are of clinical and physiological reality. These limitations are not discussed or acknowledged in the paper in any way.

*Other points of weakness in the paper:*

- There are extensive experiments performed to demonstrate the advantages of the proposed learning approach yet the discussion and presentation of the results is unfocused and difficult to follow in places. There are four experimental questions raised in the introduction and at the end of Section 4. The presented results do not address or answer these questions directly. A good place to do this would have been in a discussion section at the end of Section 5. This being said, as written, the fourth question is not really a question at all. It rather stands as a statement of what should be an objective of the paper, demonstrating how to build preference-based agents.
- Section 3 mentions the development of a platform. This however was never discussed or introduced. Typically the concept of a simulation platform denotes a standard tool or API such as OpenAI Gym or Mujoco. This was not formalized and provided a sense that the authors were trying to make unsupported claims about the paper’s significance.
- How are the environments interacted with? There are no formalized definitions of what the state or action spaces are. It is never explained how long trajectories are (ie. how many treatment decisions are possibly made for an individual patient). What are the relevant parameters of variation within the simulated dynamics? Equations for the treatment dynamics of the cancer and sepsis simulators are presented without any description of what the variables represent. How are patient physiologies varied? (patient types are mentioned in the paper) How different are the physiological responses? Do they admit different policies (aside from the trivial behavior observed in the sepsis simulator)? Provided the lack of variation in the reported results over the different data sets (namely in Figures 2, 3, 4, etc), it appears that there’s no variation nor differences between the training and validation sets. (Why wasn’t the test set evaluations presented even though it’s mentioned in the baselines?)
- The parameterizations of the reward estimators $R$ are never described. How does performance change based on the “accuracy” of these estimators?
- What does it mean to sample a patient/subject? Why is the number of patients fixed for each subset of the data? Are these numbers proxies for the number of training episodes used to learn and evaluate policies with? What does it mean to validate an RL policy when it appears policies are optimized from scratch in Figures 2, 4 and etc.?
- What is the “upper bound” presented in Figure 1?
- Why are there different time scales between Figure 2 (a) and (b)? Why isn’t expected return plotted in Figure 2(b)?
- There isn’t a clear separation between the evaluated metrics and the rewards/expected returns. Of course if one increases, the other should as well, right?


#### **General Comments**
Overall, I found this paper to be incomplete despite the extensive experiments. There are some great ideas at the root of the proposed approach to learning policies for treatment recommendation but there was far too little technical development for me to be confident about the stated contributions. There were not enough clear explanations or definitions of important and focal components of the proposed approach. Also, it’s not clear that the simulators are best suited for comparing the proposed approach with the baselines, the fact that the validation and training curves match exactly raise significant doubts that there is any variation between settings within the simulators. This being said, there are also minor concerns about the use of policy gradient approaches within a healthcare setting. Rollouts using inaccurate policies and approximate dynamics models will not be representative of real-world physiologies. Policies developed from non-physical behavior cannot be deployed in practice and are not at all reliable. While PG approaches are admissible in simulators, their suitability is extremely limited in real settings.

Ultimately, I feel that my opinion of the paper could be improved if sufficiently clear descriptions of the following concepts were provided:
- What explicit preferences were used for extracting the rewards for training AbRM and SbRM.
- Related, what are the target rewards for the learned reward functions $R$?
- What is the loss function for the agent? (Alg 3., Line 10)
- How are patient physiologies varied in the simulators? Are the differences meaningful (ie. do they provide unique policies)?



**Relevant Literature**
There were a few pieces of prior work that I expected to be included in this paper given its focus on reward design and the definition of objectives. Within the healthcare space there has been some notable work done to address reward design for RL approaches. Recently, Prasad, et al [ACM CHIL; 2020] looked into admissible reward functions for acute care.

In the IRL space, reward design has been a large area of research. Among several good papers I wanted to highlight Hadfield-Menell, et al [NeurIPS; 2017] and Shah, et al [ICML; 2019].

For Multi-objective Markov Decision Processes, I would refer the authors to Lizotte and Laber [JMLR; 2016].

Finally, Yu, et al [arxiv; 2019] put together a decent survey of RL in Healthcare that might help round out the setting of the proposed preference-based approach.


Prasad, Niranjani, Barbara Engelhardt, and Finale Doshi-Velez. "Defining admissible rewards for high-confidence policy evaluation in batch reinforcement learning." Proceedings of the ACM Conference on Health, Inference, and Learning. 2020.

Hadfield-Menell, Dylan, et al. "Inverse reward design." Advances in neural information processing systems. 2017.

Shah, Rohin, et al. "On the Feasibility of Learning, Rather than Assuming, Human Biases for Reward Inference." International Conference on Machine Learning. 2019.

Lizotte, Daniel J., and Eric B. Laber. "Multi-objective Markov decision processes for data-driven decision support." The Journal of Machine Learning Research 17.1 (2016): 7378-7405.

Yu, Chao, Jiming Liu, and Shamim Nemati. "Reinforcement learning in healthcare: A survey." arXiv preprint arXiv:1908.08796 (2019).

---

> ### Author Response · Authors · 2020-11-23
> **Explicit preferences to infer reward functions**
>
> 1. Obtaining preferences:
>
> In section 5.3, we briefly describe the preferences to consider for Cancer and Sepsis treatment. We extract the most commonly used evaluation metrics from treatment recommendation literature and use them as preferences.
>
> Overall, two policies are compared firstly based on clinical efficacy, i.e., capability to save subject’s life; if both two lead to survivals, then policy influences on subject’s health conditions are considered for a further comparison. If no policy is preferred to the other after comparison, then the two are incomparable.
>
> | Disease | Preference                                                 | First comparison          | Second Comparison                                                  |
> |---------|------------------------------------------------------------|---------------------------|--------------------------------------------------------------------|
> | Cancer  | CE                                                         | Maximizing survival rate  | -                                                                  |
> |         | CE&OF-I                                                    | Maximizing survival rate  | Minimizing sum of tumor size and toxicity level                    |
> |         | CE&OF-II                                                   | Maximizing survival rate  | Minimizing the highest toxicity level, minimizing final tumor size |
> | Sepsis  | Optimizing clinical efficacy only                          | Maximizing survival rate  | -                                                                  |
> |         | Optimizing clinical efficacy & mitigating negative impacts | Maximizing survival rate  | Reducing number of operations                                      |---------|------------------------------------------------------------|---------------------------|--------------------------------------------------------------------|
> |
>
> We elaborate how different preference concepts are utilized to compare two policies as follows:
>
> * Cancer:
>
>   * CE: Maximizing survival rate (results in main/Table 1). The policy saving the subject’s life is preferred to the policy leading to subject’s death. Otherwise, the two policies are incomparable.
>
>   * CE&OF-I: Maximizing survival rate and mitigating negative impacts in terms of minimizing sum of tumor size and toxicity level at the end of simulation (results in main/Table 1). The policy saving the subject’s life is preferred to the policy leading to subject’s death. If both lead to subject’s survival, then the policy with a smaller sum value of tumor size and toxicity level is preferred. Otherwise, the two policies are incomparable.
>
>   * CE&OF-II: Maximizing survival rate and mitigating negative impacts in terms of minimizing the highest toxicity level and final tumor size during simulation (results in appendix/Table 2). The policy saving the subject’s life is preferred to the policy leading to subject’s death. If both lead to subject’s survival, then the policy with a smaller highest toxicity level and a smaller final tumor size simultaneously is preferred. Otherwise, the two policies are incomparable.
>
> * Sepsis
>
>   * Optimizing clinical efficacy only: Maximizing survival rate (results in main/Fig.1(a)). The policy saving the subject’s life is preferred to the policy leading to subject’s death. Otherwise, the two policies are incomparable.
>
>   * Optimizing clinical efficacy & mitigating negative impacts: Maximizing survival rate and reducing number of operations (results in main/Fig.1(b)). The policy saving the subject’s life is preferred to the policy leading to subject’s death.  If both lead to subject’s survival, then the policy with fewer times of operations is preferred. Otherwise, the two policies are incomparable.
>
> 2. Utilizing preferences to infer reward:
>
> Given two policies for one sampled subject (starting with the same initial state), we can obtain the preference between the two through the above procedure. Then we can infer the reward function as Equation (1) in Section 4.2: we are minimizing the loss of a binary classification task, in which one policy is predicted with label preferred or non-preferred. We also denote the policy comparison outcome given some preference metric by EVALUATEPREFERENCE($\tau_1, \tau_2$)  in Algorithm 1.
>
> 3. Differences with handcrafted rewards:
>
> In this paper, preference-based reward is parameterized and updated based on the preference between two policies. For a given state and action, we can infer the preference-based reward based on the learned reward function (either AbRM or SbRM). So the only prior knowledge we need from human is the metric to compare two policies. In our case, the preference metric is pre-defined (e.g., the policy maximizing survival rate and reducing operation times is preferred) and the preference generation process is automatic without human’s interaction or involvement.

---

> ### Author Response · Authors · 2020-11-23
> **Loss function for the agent (Alg 3., Line 10) & How are patients physiologies varied in the simulators**
>
> 1. Loss function for the agent (Alg 3., Line 10)
>
> Since the preference-based reward is a non-stationary value approximated by a neural network, we implement agents with the policy gradient algorithm, which is robust to changes in the reward function [1, 2]. In policy gradient, we maximize the expected total reward by repeatedly estimating the gradient, and optimize it with gradient descent. Specifically, we subtract a baseline (determined by the current state only) from the expected return to avoid high variance in policy update.
>
> 2. How are patients physiologies varied in the simulators
>
> - General Cancer and drug treatment simulation
>
> We use the mathematical model proposed by [3] to simulate the general cancer evolution and drug treatment effects. There are two observations as state for each subject, i.e., tumor size and toxicity level. For state initialization, the tumor size and the toxicity level in the 0th month are generated independently from the uniform distribution $U (0, 2)$ (Section A.3.1).  Randomness also comes from the survival analysis (Section A.3.1), where the subject is likely to die intermediately.  Figure 4(b) also reflects the distinctions among patients after receiving treatment from policy gradient by observing the distribution of sum of tumor size and toxicity level (ranging from 0 to more than 5) at the end of simulation from x-axis. Similar distributions can be observed in Figure 4(c) for receiving treatments recommended by SbRM, 7(b) for receiving treatments recommended by AbRM. To further visualize the subject differences, we add one more plot in Appendix Figure 11 (also see in the link: https://drive.google.com/file/d/1actrpcPgtNr5CVrhBndB5RjCddUCCrz-/view?usp=sharing ), in order to show the value distribution of tumor size and toxicity level along simulation, either without or with treatment recommended by all 12 compared approaches.
>
> - Sepsis infection and blood purification simulation
>
> We employ the mathematical model derived by [4] to simulate the acute inflammation process in response to an infection. We have eight longitudinal measurements of key cytokines and damage-related markers in blood of  23 CLP-induced septic rats. There are 19 physiological features that govern sepsis dynamics, 8 of which are observable while the remaining 11 are unmeasurable conceptual variables. All the state features are initialized by 0 except $CLP$ (0.9, cecal ligation and puncture) and $N_r$ (2.5e6, resting blood neutrophils). The parameters in the state transition functions (21 ODEs) characterizes the subject, and we assume they follow normal distribution with respective mean and standard deviation known from existing literature. We sample subjects individually by generating ODE parameters using Markov-Chain Monte Carlo (MCMC) sampling of their posterior parameter distributions. The parameters for each subject are accepted if they are compatible with their respective distributions and the observations in the resulted simulation are close to experimental data (measurements of 23 rats). Therefore, the transition parameters that distinguish one subject from the other should be quite different from each other due to the random initialization (follow their gaussian distributions) and iterative update during the MCMC sampling process. To further visualize the subject differences, we add one more plot in Appendix Figure 13 (also see in the link: https://drive.google.com/file/d/1HNY_Soo0r4fhcbQAx_mc5tYMC_XslS4A/view?usp=sharing ), in order to illustrate the variation of 19 state features during the simulation without treatment or with treatment recommended by SbRM.
>
>
> [1] Ho, Jonathan, and Stefano Ermon. "Generative adversarial imitation learning." Advances in neural information processing systems. 2016.
>
> [2] Christiano, Paul F., et al. "Deep reinforcement learning from human preferences." Advances in Neural Information Processing Systems. 2017.
>
> [3] Zhao, Yufan, Michael R. Kosorok, and Donglin Zeng. "Reinforcement learning design for cancer clinical trials." Statistics in medicine 28.26 (2009): 3294-3315.
>
> [4] Song, Sang OK, et al. "Ensemble models of neutrophil trafficking in severe sepsis." PLoS Comput Biol 8.3 (2012): e1002422.

---

> ### Author Response · Authors · 2020-11-23
> **Some minor points to clarify**
>
> **Observed gains with AbRM and SbRM**: Since two agents with different parameter initializations are used in the PRL framework, so for fair comparison, we also have a compared method called Single-objective RL (Ensemble) in Table 1 for Cancer and RL (Ensemble) in Figure 1, where two agents are trained separately and the one with better performance on the validation set is evaluated on the testing set.
>
> **Off-policy settings**: learning policies from simulation and from observations are two branches of approaches to perform treatment recommendations. We also conduct additional experiments about off-policy learning and evaluation on Cancer treatment recommendation: Given trajectory histories of a policy gradient learner with 26.96% survival rate (behavioral), we train another DQN learner without interactions with the simulator (learner). We then test the learner in the simulator and find that the achieved survival rate has dropped to 16.12%. We further illustrate the inaccuracy of existing importance sampling approaches for off-policy evaluation in Appendix Fig. 10 (also see in the link: https://drive.google.com/file/d/1T2OA4QG9bvlSYgVkRsQExchkk_WhrGRU/view?usp=sharing ).  When testing the well-trained agents on simulators, the actual expected return mainly focuses on −1 since extremely low survival rate is achieved by agents learning policies from other policies. However, the expected returns estimated by Per-decision Importance Sampling (PDIS), Weighted Per-decision Importance Sampling (WPDIS), Doubly-robust (DR) and Weighted Doubly-robust(WDR) distribute quite differently from the ground-truth. Both of the above unreliable off-policy evaluations and inferior treatment outcomes resulted from policy learning based on observational data demonstrate the importance of the provided simulation platform in enabling optimal policy learning and ensuring accurate policy evaluations.
>
> **Simulation platform**: Typically, the agent receives the current time and state from the simulator, and then sends the action back. We will upload the simulator codes and document later. There is no particular domain shift between training and testing set, but distinctions exist among different sampled subjects. Some details about the simulator are as follows:
> * Cancer: there are two state features: tumor size and toxicity level; there are four possible actions to take: 0.1, 0.4, 0.7, 1.0 (representing the dosage amount). One subject experiences at most 6 time-step simulations if he can survive to the end, each representing one month in the real-world. State transition dynamics are formulated in Section 3.1.
> * Sepsis: there are 19 physiological state features, 8 of which are observable while the remaining 11 are unmeasurable conceptual variables; there are two possible actions to take, one is performing blood purification operations and the other is no operation. One subject experiences 1000 time-step simulation, each representing 0.1 h in real-world. There are 21 ODEs to represent state transitions, some of them are important and have been listed in Section 3.2 and A.3.2. The interaction network of inflammatory responses and hypothetic hemoadsorption mechanisms is also demonstrated in Figure 12 (Appendix).
>
> **Parameterization**: both agent and reward learner are parameterized by two-layer MLPs for Cancer (MDP) and two-layer LSTMs for Sepsis (POMDP). Figure 3 can reflect the relationship between reward estimator accuracy and policy performance. While preserving the incomparable policy for reward estimator (blue curves), the preference over incomparable policy becomes stable around 10th epoch (Fig.3 (a)), till which the policy learner has steady increasing survival rate (Fig.3(c)); after that the accuracy of reward estimator does not change, but the policy learner keeps on improving policy until it converges.
>
> **Sample a patient**: for both Cancer and Sepsis experiments, we can generate a random subject (initial tumor size and toxicity level for Cancer and state transition parameters for Sepsis). We set the training and testing set fixed for fair comparison. A large number of training set is beneficial for learning parameters in deep models and a large number of testing set is appropriate to evaluate the performance with small variance. Validation at different epochs during model training is designed for selecting the epoch with best validation results for testing.
>
> **Different time scales in Fig.2**: Training with or without a pre-trained reward model will converge at 10th epoch, so we only show the first 12 epochs to illustrate the performance gap between training with transfer and training without transfer.
>
> **Evaluation metrics and rewards/expected return**: Actually, the evaluation metric is the preference metric that guides the preference generation between two policies. Since the reward is inferred rather than pre-defined, then the two should have a positive proportional relationship if the reward estimator is well-trained.

---

### Decision · Program_Chairs · 2021-01-07
**Final Decision**

**Decision:**

Reject

**Comment:**

Reviewers agree that this work is promising. The paper is well-grounded in the literature and different aspects of the considered methods are investigated through a variety of experiments. Unfortunately, this paper does not provide sufficient details to allow the reader to understand what has been done nor how to adequately build from it. For example, details in the Appendix lack sufficient formalization of the equations or concepts used to train the preference-based agents. The paper would benefit from clarifications of the method, procedures, and equations used. Beyond that, a major concern lied within the evaluation of the simulated patients across different initializations. Provided that one of the proposed contributions of this paper is a robust simulation platform for RL research within healthcare, it would be important to report convincing results on the patient physiologies admitted by the simulator and characterizing the behaviors of policies learned using this simulator. Finally, issues regarding the structure of the paper, including the split between the main paper and the Appendix, should be resolved before this paper can be published. Notably, the authors should consider elevating important material from the Appendix into the main paper.